# The Harvard USPTO Patent Dataset:
# A Large-Scale, Well-Structured, and Multi-Purpose Corpus of Patent Applications

**Mirac Suzgun[a]**   **Luke Melas-Kyriazi[b]**   **Suproteem K. Sarkar[c]**
**Scott Duke Kominers[c,d]**   **Stuart M. Shieber[c]**
[a]Stanford University   [b]Oxford University   [c]Harvard University   [d]a16z crypto

[*]Correspondence to: msuzgun@cs.stanford.edu

## Abstract

In this paper, we introduce the Harvard USPTO Patent Dataset (HUPD), a large-scale, well-structured, and multi-purpose corpus of English-language patent applications filed to the United States Patent and Trademark Office (USPTO) between 2004 and 2018. With more than 4.5 million patent documents, HUPD is two to three times larger than comparable corpora. Unlike previously proposed patent datasets, HUPD contains the inventor-submitted versions of patent applications—not the final versions of granted patents—thereby allowing us to study patentability at the time of filing using NLP methods for the first time. It is also novel among NLP datasets in its inclusion of rich structured metadata alongside the text of patent filings, enabling researchers to perform new NLP tasks leveraging this metadata. As a case study on the types of research HUPD makes possible, we introduce a new task to the NLP community—patent acceptance prediction. Finally, we demonstrate how our dataset can be used for three additional tasks: multi-class classification of patent subject areas, language modeling, and summarization. Overall, HUPD is one of the largest multi-purpose NLP datasets containing domain-specific textual data, along with well-structured bibliographic metadata, and aims to advance research extending language and classification models to diverse and dynamic real-world data distributions.[1]

## 1   Introduction

Patents are key public indicators of innovation and technological advancement. They offer a simple yet powerful source for studying, measuring, and appraising innovation activity, economic growth, and emerging technology. Over the past two decades, the total number of patent applications filed to the United States Patent and Trademark Office (USPTO) per year has almost doubled. In the fiscal year 2020 alone, the USPTO received more than 650,000 patent filings, including requests for continued examinations [1]. The competitive and regulatory landscape surrounding patent-driven innovation is rapidly evolving, but despite the clear focus in textual data in the field of patent analysis, it has yet to be systematically studied by the ML and NLP communities.

The absence of large-scale, well-structured, and distilled patent data is a major hurdle preventing researchers and practitioners from applying ML tools to understand and explore innovation and technological change through patent text. In recent years, there have been efforts to produce NLP

---

[1]Our dataset, along with our code and models, is publicly available at https://patentdataset.org and distributed through HuggingFace Datasets (https://huggingface.co/datasets/HUPD/hupd). All the authors were based at Harvard University at the inception of this project.

37th Conference on Neural Information Processing Systems (NeurIPS 2023) Track on Datasets and Benchmarks.

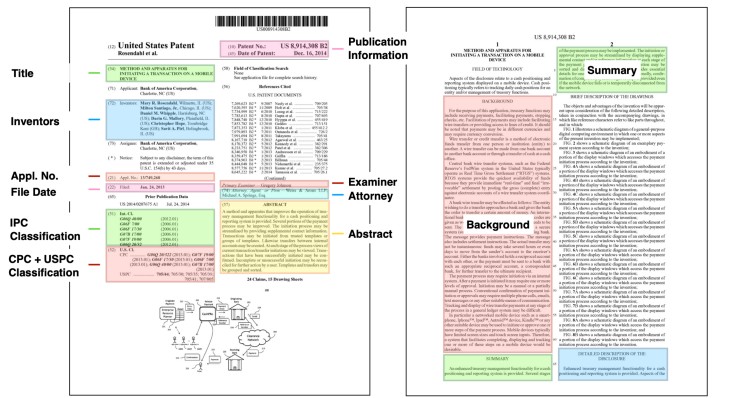
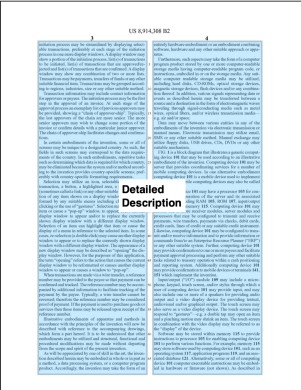

**Figure 1:** Three pages of the pre-grant version of an example patent document (*Method and Apparatus for Initiating a Transaction on a Mobile Device* [Publication No: 2014-0207675 A1]). The highlighted sections show a subset of the 34 data fields that we include in the Harvard USPTO Patent Dataset.

datasets of patent text, including CLEF-IP 2011 [2], USPTO-2M [3], and BIGPATENT [4]. These datasets have some limitations in their scopes and features, as shown in Table 1. They may contain text and information only for granted patents, may contain subsets of patent text and metadata, and typically focus on only one particular NLP task.[2]

It is thus highly desirable to have a free, publicly-available dataset that provides a wider and more encompassing repository of patent data—covering multiple sections and years—of not just granted patents but all patent applications, that allows more flexibility and control in data selection, and that can be appropriated for multiple experiments and investigations. With these desiderata in mind, we introduce a public, large-scale, consistently-structured, and multi-purpose corpus of patent data to the NLP community, called the Harvard USPTO Patent Dataset (HUPD). The dataset contains more than 4.5 million English-language utility patent applications filed to the USPTO between 2004 and 2018, and aims to advance research efforts in both patent analysis and NLP.

HUPD distinguishes itself from prior NLP patent datasets in three key aspects. First, unlike some other datasets, it does not restrict the sample to only granted patents, and instead focuses on patent applications. Patent applications contain the original set of claims and descriptions of the proposed invention written by the applicant. Because the dataset has a consistent set of patent documents at the time of filing, it avoids the dataset shift concerns that would be present in a study of accepted and rejected patent applications at different revision stages. In fact, having access to the original versions of both accepted and rejected applications allows us to introduce a completely new task to the field—binary classification of patent decisions. The goal of this task is to predict the patent acceptance from the language of a patent application at the time of submission. Second, our dataset features multiple classes of rich textual and structural information present in patent applications. Whereas many previous NLP datasets include only one or two of a patent's data fields (for example, description and abstract), HUPD contains 34 fields, including filing date, fine-grained classification codes, examiner information, and many others. The variety of information available for each patent application can enable NLP researchers to perform a wide range of tasks—such as analyzing the evolution of patent language and categories over time—that were not possible under previous NLP patent datasets. Third, HUPD uses textual and bibliographic information obtained directly from the USPTO's data products, rather than from Google's Patent search as BIGPATENT does; it is larger than previous NLP patent datasets while still being clean, comprehensive, and well-structured.

We introduce our patent dataset with several audiences in mind. For the NLP community, the universe of patent applications—with its sheer breadth and wealth of textual substance and structured format—provides an ideal domain-specific laboratory for developing and evaluating new NLP tools.[3] From

---

[2]We include a more detailed discussion of these NLP datasets, as well as other popular raw patent document repositories and search tools not produced primarily for the NLP community, in Section C.

[3]Patent applications follow a strict and structured framework: They generally contain claims, description, drawings, background, abstract, and summary sections. It is imperative—for both patent prosecution and any potential future litigation issues—to define the scope of the claims of the proposed invention clearly and completely, and to avoid misleading or confusing statements (for more information, see 35 US Code

| Dataset | # Docs | Title | Abst | Appl | Exam | Invt | PD | Claims | Bkgd | Dsc | PCs | Years | Primary Purpose |
|---|---|---|---|---|---|---|---|---|---|---|---|---|---|
| WIPO-alpha | 75,250 | ✓ | ✓ | | ✓ | ✓ | ✓ | ✓ | ✓ | ✓ | ✓ | 1998-2002 | Classification |
| CLEF-IP (2011) | 1,500,000 | ✓ | ✓ | | | ✓ | ✓ | ✓ | | ✓ | ✓ | < 2009 | Retrieval+Classification |
| USPTO-2M | 2,000,147 | ✓ | ✓ | | | | | ✓ | | | ✓ | 2006-2015 | Classification |
| BigPatent | 1,341,362 | ✓ | ✓ | | | | | | | ✓ | | 1971-2018 | Summarization |
| **Ours (HUPD)** | **4,518,263** | ✓ | ✓ | ✓ | ✓ | ✓ | ✓ | ✓ | ✓ | ✓ | ✓ | **2004-2018** | **Multi-Purpose** |

**Table 1:** Comparison of HUPD with other datasets whose primary goal is NLP patent analysis. The abbreviated columns mean the following. *Abst*: Abstract, *Appl*: Applicant Information, *Exam*: Examiner Information, *Invt*: Inventor Information, *PD*: Publication Date, *Bkgd*: Background, *Dsc*: Description, and *PCs*: IPC/CPC codes.

abstractive summarization of patent sections to information retrieval and named-entity recognition and extraction, one can perform a variety of both standard and novel NLP tasks on patent data. Furthermore, the well-structured nature of our dataset allows for researchers to study how concepts like acceptance criteria vary across contexts and over time. For the IP community, a key motivator of our work is that there are many rote tasks associated with patent filing and examination—including the categorization of patents into relevant technology areas and prior art search—in which machine learning methods could potentially be used to provide efficiency, value, and cost savings. Finally, for the general audience, our present work illustrates how NLP/ML tools may be efficiently deployed for advancing socially relevant objectives and studying diverse and dynamic application areas.

## 2 Preliminaries and Background

A patent application[4] typically consists of a title, abstract, set of claims, detailed description, drawings (if needed to describe the invention), and cross-references to related applications, among other written specifications. Applications are filed to the USPTO and reviewed by examiners who are expected to have knowledge and expertise associated with the subject matter of the invention.

During the examination process, an examiner determines the *patentability* of the invention. The examiner decides whether the proposed invention is *useful*, *non-obvious*, and *statutory*, and searches for already-existing patents within the technology sphere of the invention to confirm the set of claims provided is *novel*. Afterwards, the examiner sends an Office action to the applicant, notifying them of the USPTO's decision. If the decision is favorable, then the applicant can choose to proceed with their application and have the USPTO issue their patent. However, if the decision is unfavorable, then the applicant receives a notification of rejection; it is then up to the applicant to decide whether they wish to respond to the rejection, continue to pursue their application, and request a reexamination.

It is common to think of patent applications as being simply accepted or rejected, but the status of an application is considerably more subtle. For the task of predicting an application's outcome, it is useful to provide a formalization of what it means for a patent application to be *accepted* and *rejected*. We say that a patent application is "accepted" if it has been officially approved, granted, and published by the USPTO. There is, however, no clear-cut notion of absolute *rejection* in patent applications. An application might, and in fact often does (at the beginning), receive an Office action indicating a *non-final* or *final rejection*—typically on the grounds of prior art, scope of the claims, or lack of utility or novelty or obviousness, but the applicant can submit a response to the USPTO and re-open the prosecution of their application, even in the case of a final rejection. As a result, a "final rejection" does not imply a patent application has no chance of eventually being accepted after revisions. In our dataset, we label an application "rejected" if it has received an Office action of rejection—final or non-final—and was ultimately abandoned by the applicant.[5] We categorize all the remaining applications, which are still waiting a response from the USPTO, as "pending." These

---

§112). Conversely, patent applicants sometimes make strategically ambiguous and creative specifications in their applications towards the goal of broadening claim coverage. These characteristics, along with other idiosyncrasies, make the patent domain a uniquely challenging and powerful setting to study through ML tools.

[4]Following the terminology used by Toole et al. [5], we define a "patent application" as a non-provisional application for an invention and a "patent" as a granted patent. Prior to 2001, patent applications to the USPTO were not published: They were instead kept in secrecy until their patent issue dates (p. 174; Menell et al. [6]). Under the American Inventors Protection Act of 1999, however, most patent applications are published eighteen months after the earliest filing date, unless the applicant requests earlier publication.

[5]Following the "Manual of Patent Examining Procedure," we treat a patent application as "abandoned" if its applicant either fails to take appropriate action and to reply to the USPTO within a specific time period or submits a written declaration of abandonment. In the case of non-final rejections, for instance, applicants are usually given six months to revise their applications [7].

distinctions are particularly useful when describing and discussing the binary classification task of patent decisions in the following sections.[6]

# 3    Related Work on Patent Analysis

Existing patent datasets for NLP focus nearly exclusively on two tasks: patent subject classification and summarization. Below, we provide overview of prior datasets and studies in these areas.[7,8]

**Automated Subject Classification.** Patents are classified by subject matter according to standard taxonomies, most notably the International Patent Classification (IPC) and Cooperative Patent Classification (CPC) systems.[9] These IPC/CPC codes are hierarchical—classified at a class level (e.g., G-*Physics*), subclass level (e.g., G06F-*Electric Digital Data Processing*), and so on. Previous studies attempted to predict the IPC or CPC codes of patents at the class and subclass levels using various statistical methods, including classical statistical learning tools [12–16] and neural models [17, 3, 18]. Recently, Transformers [19] have been considered for this task: Lee and Hsiang [20], for instance, fine-tuned a pre-trained BERT [21] to predict IPC/CPC codes of patents. Zaheer et al. [22] conducted similar experiments using BIGBIRD [23] and showed improvements over BERT.[10]

As shown in Table 1, WIPO-alpha, CLEF-IP, and USPTO-2M have been the main gymnasia for model training for the IPC/CPC classification tasks, but these corpora are still limited in their scopes. They contain a relatively smaller set of patent text and metadata, do not allow users to choose which year ranges to focus on during training and testing, and in some cases come pre-tokenized, which may prevent users from using custom vocabularies. HUPD addresses these limitations, and also allows more flexibility in data selection and provides more comprehensive text, field, and year coverage.

**Patent Text Generation and Summarization.** With the growing availability and success of large language models in recent years, there has been an interest in applying language models to patents. Sharma et al. [4] initiated such explorations, introducing the first summarization dataset on patents, called BIGPATENT, and trained summarization tools on their dataset to generate the abstract section of a patent given its description section. The BIGPATENT dataset contains 1.3 million utility patents filed to the USPTO between 1971 and 2018, and was collected from the Google Patents Public Dataset via BigQuery. Our dataset differentiates itself from BIGPATENT in three aspects: (1) HUPD includes metadata and fields, including claims, background, filing date, and examiner information, that are not present in BIGPATENT; (2) in addition to accepted patents, it contains rejected and pending applications—it thus enables the study of patent acceptance/rejection[11]; and (3) it has approximately three times as many documents as BIGPATENT (Table 1).[12]

**Patent Acceptance Prediction.** To the best of our knowledge, our study is the first work to introduce a practical definition of rejection in patent examination to identify, analyze, and discuss the patterns in and characteristics of accepted versus rejected patent applications from a *purely textual* perspective. In that sense, we introduce the patent decision classification task to the NLP literature.

---

[6]We refer our readers to [8] for a discussion on data controversies around calculating patent grant rates.

[7]In Section C, we provide a more thorough and comprehensive comparison of our dataset against BIGPATENT and other existing large-scale NLP datasets (such as S2ORC [9] and WikiBio [10]).

[8]For an alternative compendium of studies that make use of deep learning techniques for patent analysis, please see the study by Krestel et al. [11].

[9]Established by the Strasbourg Agreement of 1971 and administered by the WIPO, the IPC scheme is currently being used, in various capacities, in over one hundred major patent offices worldwide. The CPC scheme is an extension of the IPC scheme and has been adopted by the USPTO since 2013. It offers a more comprehensive coverage of some new technical developments and includes an additional section "Y." In the US, every issued patent needs to be classified, based on its subject matter, according to the CPC taxonomy.

[10]In addition, Acikalin et al. [24] classify the scope of a patent in terms of exposure to the U.S. Supreme Court decision *Alice Corp. v. CLS Bank International*, 573 U.S. 208 (2014).

[11]Furthermore, HUPD contains the original, inventor-submitted version of patent applications, as opposed to the granted versions. The distribution of language in the granted versions of accepted patents may change after examiner comments, making granted versions less directly comparable to pending patent applications. For those patent applications that are eventually granted and published, the published version can be retrieved using the patent number metadata in our dataset.

[12]We include a more in-depth comparison of our dataset with BIGPATENT in Appendix C.

| Section | Brief Description | Avg # Tokens |
|---|---|---|
| Title | Title of the Invention | 16.4 |
| Abstract | Summary of the Background and Claims | 132.0 |
| Claims | List of Items Defining the Invention | 1271.5 |
| Background | Brief Statement of the Field of Art and Related Art of the Invention | 627.11 |
| Summary | Condensed Version of the Description | 917.8 |
| Description | Detailed Statement and Disclosure of the Invention | 11855.6 |

**Table 2:** Brief description of and average number of tokens in each text-based section in HUPD, as measured by the GPT-2 tokenizer. Typically, the description section in a patent application is almost 100 times longer than the abstract section. HUPD can be used for long-sequence summarization and language modeling, inter alia.

## 4 The Dataset

The Harvard USPTO Patent Dataset (HUPD) contains 4,518,263 utility patent applications filed to the USPTO between January 2004 and December 2018.[13] In this section, we elaborate on the data collection process and provide details about the data format and dataset structure. We furthermore enumerate and highlight a portion of the data's statistical properties, and acknowledge the limitations and ethical considerations of the present work. Additional information is included in Section A.

**Dataset Construction.** As specified by US law, all patent data is publicly accessible. Distilling this information into a NLP dataset involves obtaining the patent data and its corresponding metadata, normalizing all data to the same format, filtering missing and erroneous data, de-duplicating data, and merging all data into a single easy-to-use dataset. Here we provide an overview of this process.

Patent application texts were obtained from the USPTO Bulk Data Storage System (BDSS; Patent Application Data/XML Version) as XML files.[14] As not all the original patent files follow the same XML structure, we wrote regular expressions to parse all the different formats into a normalized set of data fields, and stored these data fields as structured JSON files. Filing metadata—including acceptance decisions, filing dates, titles, and classification information—were separately obtained from the USPTO Patent Examination Research Datasets [26, 27] in February 2021. This metadata was then merged with the full-text patents from the USPTO BDSS to link patents filed before 2020 with information about examiners, office actions, and additional filing information. This merging process ensures that our dataset contains both the updated metadata structure and the patent application texts.

Finally, we assembled the continuation information for each application as follows: Applications with parent filings in the USPTO continuity data files were marked with the prefix "CONT-" in the decision status field.[15] Hence, there are six labels in total for the decision status of applications: "Accepted," "Rejected," "Pending," "CONT-Accepted," "CONT-Rejected," and "CONT-Pending."[16]

**Statistics.** Table 1 provides a quantitative comparison of our dataset with other patent data sources, while Table 2 gives information and statistics about the text-based sections in our dataset. HUPD builds on its counterparts, including BIGPATENT, because of not only its size and wider coverage but also its ability to allow users to see the evolution of patent families over time.

**Limitations.** From a methodological point of view, the dataset is limited to patents from the U.S. and in English, and drops image content such as drawings. From a functional point of view, some textual sections are longer than current NLP models can process, and specialized vocabulary in certain fields can create issues for existing tokenizers.

---

[13]We chose to restrict our attention to utility patent applications both because utility patent applications are treated as "inventions" and because they have constituted more than 90% of total patent applications every year in the past two decades (and beyond) [25]. Our dataset does not include any design or plant patent submissions.

[14]Per the USPTO's Electronic Information Products Division's Terms and Conditions, bulk data products are provided with no restrictions on use.

[15]Continuity data in patent applications establishes a link between an initial patent application and its subsequent continuation applications. This information is accessible on the PatEx website through two distinct files. The "continuity parents" file contains details about the original parent application, and the "continuity children" file includes information on the subsequent continuation applications derived from the parent. Collectively, these two files are referred to as the USPTO continuity files in the context of this paper. We labeled all continuations that were not continuing a provisional application or national stage entries as "CONT-."

[16]To avoid duplicate documents in our corpus, we excluded CONT-applications from our experiments; however, users may choose to use them to study changes in patent applications for the same invention over time.

| Task Name | Setup | Metrics |
| --- | --- | --- |
| Acceptance Prediction | Abstract or Claims $\rightarrow$ Patent Outcome (Decision) | Accuracy |
| IPC/CPC Classification | Abstract or Claims $\rightarrow$ IPC/CPC | TOP-$k$ |
| Language Modeling | Abstract or Claims or Description | Perplexity |
| Abstractive Summarization | Claims or Description $\rightarrow$ Abstract | ROUGE/BLEU |

**Table 3:** Summary of the four NLP tasks presented in this work, along with some evaluation metrics for them. Our dataset can be used to conduct many other NLP/IP experiments. See Section 5 for detailed information.

**Potential Biases.** We provide a detailed examination of HUPD for potential biases in the Appendix. We recommend that researchers who use HUPD consider these biases when interpreting their results—especially if their analyses involve inventor and/or examiner information. For now, we summarize our empirical findings in this section and note they are consistent with the previous work [28–32]. We show, among other results, that female inventors are notably underrepresented in the U.S. patenting system, that small and micro entities (e.g., independent inventors, small companies, non-profit organizations) are less likely to have positive outcomes in patent obtaining than large entities (businesses with more than 500 employees), and that patent filing and acceptance rates are not uniformly distributed across the US. Our empirical findings suggest that any study focusing on the acceptance prediction task, especially if it is using the inventor information or the small-entity indicator as part of the input, should be aware of the the potential biases present in the dataset and interpret their results in light of these considerations.

**Ethical Considerations.** While building HUPD, we followed the data sheets and statements introduced by Gebru et al. [33] and Bender and Friedman [34], discussing the motivations, objectives, collection process, workflow, use cases, distribution, maintenance, potential contributions, and potential misuses of our dataset and research (see: Section A).[17]

**Impact on Underserved Communities.** HUPD contains patent applications in English, a language with heavy attention from the NLP community. However, innovation is spread across many languages, cultures, and communities that are not reflected in this dataset. HUPD is thus not representative of all kinds of innovation. Furthermore, patent applications require a fixed cost to file and are not accessible to everyone. One goal of our dataset is to spur research that reduces the cost of drafting applications, potentially allowing for more people to seek intellectual property protection for their innovations.

## 5 Using the Dataset

Due to the versatile nature of HUPD, a wide range of NLP tasks and experiments can be constructed from the dataset by selecting the appropriate fields and metadata contained in each patent application. In this section, we highlight four tasks that we believe to be among the most valuable and relevant to the NLP and IP communities: (i) binary classification of patent decisions, (ii) multi-label classification of patent IPC/CPC categories, (iii) language modeling, and (iv) summarization. Of course, the dataset may easily be used to conduct other investigations, such as patent clustering, prior art search, and early detection of superstar inventions.[18] In what follows, we describe our four tasks of interest in detail and contextualize their importance within the world of IP. Table 3 provides a brief overview of each task and the corresponding metrics used to measure task performance.

**Patent Acceptance Prediction.** Given a section of an application (in particular, the abstract, claims, or description), we predict whether the application will be accepted by the USPTO. From the perspective of the NLP community, this is a standard classification task. Yet, the potential applications and benefits of this decision task, as well as its difficulty, distinguish it from prevalent binary classification benchmarks (e.g., SST, Yelp). In our experiments, we focus on applications without parent filings to make our setup simple and clear, thereby excluding all the CONT-applications. Also, we do not include any pending applications.

**Automated Subject Classification.** The next task is to predict the primary IPC or CPC code of a patent application given a subset of the text of the application. A coherent automated classification of patent documents into different technology fields can facilitate effective assignment of patent

---

[17]In Section A, we share a data card for our dataset, in the style of Gehrmann et al. [35].

[18]Since it would be challenging to simultaneously study all of these tasks made possible by our dataset, we leave some of these experiments for future work.

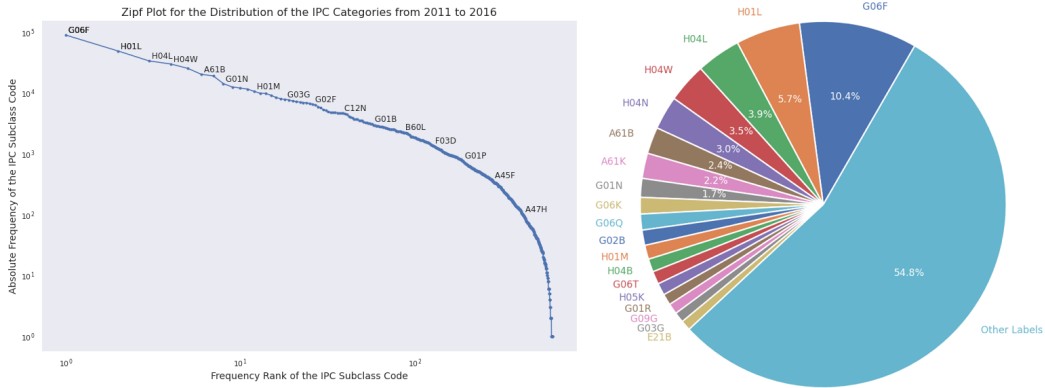

**Figure 2:** IPC distribution of accepted patent applications from 2011 to 2016 at the IPC subclass level. There are 637 IPC subclass labels in HUPD, of which the most common 20 codes make up half of the distribution. G06F-*Electric Digital Data Processing* is the largest IPC subclass, accounting for 10.4% of applications.

applications to examiners. In addition, it may help create a rigorous and standardized catalog of prior art for research and exploration. This task might also help the early identification of valuable inventions that bridge multiple technological domains or the emergence of new subject areas. In our experimental setups, we predict the main IPC codes of patent applications at the subclass level, as IPC codes are available for a larger set of patents than CPC codes.[19] There are 637 IPC codes at the subclass level in our dataset, but they are not uniformly distributed (see Figure 2); for instance, G06F-*Electric Digital Data Processing* constitutes 10.4% of accepted patents that were filed between 2011 and 2016 and the most popular 15 IPC codes make up almost 40%. Hence, it is difficult to achieve strong classification performance by only predicting the major classes.

**Language Modeling.** Next, we move from classification to language modeling (LM). We consider masked LM of patent claims. We concentrate on the claims section because it forms the basis of the invention described in a patent application; it has a distinctive language style; and it is considered to be the most legally potent part of a patent. We also conduct LM experiments on the abstract sections of patents, which are more similar to standard natural language. These models can be used for downstream tasks, as well as for domain-specific investigations. In Section H, we demonstrate one application of our LMs by visualizing the average embeddings of different patent categories under this fine-tuned model. The results of this exercise may reveal the textual evolution of innovation concepts and trends in patent applications across different technology areas.[20]

**Abstractive Summarization.** The formulation of the final task follows naturally from the structure of our patent data: Each patent contains an abstract in which the applicant summarizes the content of the patent. We use this section as the ground truth for our abstractive summarization task, and we use either the claims or the description as the source text. We perform this conditional generation task with the same motivation that inspired Sharma et al. [4]; our setup is similar to theirs apart from the size and scope of our dataset. One difference in our current setup is that our dataset allows us to explore using either the claims or description section for the source text, whereas in Sharma et al. [4] only the description section is available.[21]

## 6 Results and Discussion

In the next two sections, we establish benchmarks for the four tasks, describe the models used, and analyze our findings. In our experiments, we typically use the abstract or claims sections as the input

---

[19]Of the existing patent applications containing CPC codes (between 2011 and 2016) in our corpus, the overlap between their IPC and CPC codes at the subclass level is 99.6%.

[20]We perform masked LM, as in BERT [21], rather than autoregressive LM, in an effort to produce higher quality contextual representations.

[21]The claims sections of patent applications are typically long (on average 1272 words—almost an order of magnitude longer than the abstract sections (see Table 2)); hence, this task is inherently difficult. In contrast to most other summarization tasks (e.g., CNN/DM article summarization), it is not sufficient for the model to draw from the first or last few sentences of the input to generate the summary; rather, the information needed to generate the text may be distributed throughout the input.

| IPC – Section | BernNB | MultiNB | Logistic | CNN | DistilBERT[FT] | BERT[FT] | RoBERTa[FT] |
|---|---|---|---|---|---|---|---|
| **G06F** – *Abstract* | 61.86 | 61.47 | 58.24 | 60.97 | **61.53** | 61.28 | 61.31 |
| **G06F** – *Claims* | **63.96** | 62.06 | 58.02 | 63.38 | 63.37 | 62.97 | 63.25 |
| **H01L** – *Abstract* | 58.98 | 59.05 | 58.54 | 60.71 | 61.46 | 61.85 | **61.85** |
| **H01L** – *Claims* | 60.97 | 60.29 | 59.53 | **62.63** | 62.50 | 61.61 | 61.94 |
| **H04L** – *Abstract* | 59.35 | 58.75 | 58.75 | 59.89 | **60.54** | 60.52 | 60.05 |
| **H04L** – *Claims* | 62.13 | 61.04 | 58.04 | **62.34** | 61.42 | 61.47 | 61.74 |
| **H04N** – *Abstract* | 60.74 | 60.64 | 58.79 | 60.37 | **62.01** | 61.93 | 61.51 |
| **H04N** – *Claims* | 62.51 | 61.01 | 57.53 | **63.98** | 62.82 | 61.98 | 62.14 |
| **A61B** – *Abstract* | 59.15 | 58.81 | 57.31 | 58.75 | 58.36 | 59.58 | **59.66** |
| **A61B** – *Claims* | 59.30 | 59.12 | 57.25 | 59.49 | 60.15 | **61.20** | 61.00 |
| **G01N** – *Abstract* | 59.85 | 59.89 | 57.25 | 59.98 | 59.00 | 60.30 | **61.10** |
| **G01N** – *Claims* | 58.06 | 57.97 | 58.37 | 59.80 | 60.16 | 60.34 | **60.97** |
| **GO6Q** – *Abstract* | 61.53 | **61.64** | 58.52 | 60.46 | 61.23 | 61.09 | 61.56 |
| **GO6Q** – *Claims* | **63.96** | 63.31 | 57.17 | 62.90 | 61.88 | 62.19 | 63.25 |

**Table 4:** Baseline performances of our models on the binary classification of patent acceptance task. All the models were trained and evaluated on the patent applications filed to the USPTO between January 2011 and December 2016. All the test sets contained equal numbers of accepted and rejected applications, so the baseline accuracy to compare these models against is $50\%$. In all but one IPC category, the models trained on the claims sections yielded the best performance. None of the individual accuracy scores, however, went beyond $64\%$. In Section 7, we further report results of our conditional universal acceptance prediction classifier. The superscript [FT] denotes that these models were fine-tuned. The common IPC categories presented in this table are G06F-*Electric Digital Data Processing*, H01L-*Semiconductor Devices*, H04L-*Transmission of Digital Information*, H04N-*Pictorial Communication*, A61B-*Diagnosis, Surgery, Identification*, G01N-*Investigating or Analyzing Materials*, and G06Q-*Data Processing Systems or Methods*. (See Table 6 for our full results.)

to the models; however, future studies using our dataset can easily include other sections, such as the background and description.[22]

While our dataset includes patent applications filed to the USPTO between 2004 and 2018, we primarily focused on the 2011-2016 year range in our main NLP experiments. We chose this specific time period not only because the patent applications during these years reflect more recent inventions and cover diverse industries, but also because they were suitable for training various NLP models within a day at the time of our experiments. We used the same subset across most of our experiments for consistency and coherency. (For additional details on our experimental setup, please see: Section F).

**Patent Acceptance Prediction.** Table 4 reports the performances of our models on the most popular IPC codes. In each IPC subclass, with the exception of G01N-*Investigating or Analyzing Materials by Determining Their Chemical or Physical Properties*, the best performance was achieved by the models that relied on the claims. This result is consistent with the idea that the claims define the overall scope, novelty, and usefulness of the invention. The claims, together with the prior art, provide the most useful and critical information about the *patentability* of an invention. It was, however, surprising to discover that there was not a significant difference between the NB classifiers and the BERT models in terms of accuracy scores across categories. We speculate that the BERT models might have mirrored the behaviors of the NB classifiers at the end, failing to go beyond word-level feature extraction. Nonetheless, we note the task is difficult, and currently uses only a limited portion of a patent text to determine the acceptability of the invention. We posit that new advances in Transformer models for longer text may improve predictive accuracy for this task.[23]

**Automated Subject Classification.** For the multi-class classification task, we considered only accepted patents, since they contain reliable and accredited IPC/CPC codes. Table 5 details the TOP1 and TOP5 accuracy scores for the multi-class IPC code classification results at the subclass level. First, we note that performance increases with more sophisticated models. The DistilBERT model,

---

[22]Due to space limitations, we present a subset of our results. Additional findings are in the Appendix.

[23]The method used here is purely on internal textual features, not on past patenting history or prior art information, which are important considerations in predicting acceptance of a patent application. We leave the incorporation of such metadata into this task for future work.

| Section | BernNB | | MultiNB | | CNN | | DistilBERT [FT] | | RoBERTa [FT] | |
|---|---|---|---|---|---|---|---|---|---|---|
| | TOP1 | TOP5 | TOP1 | TOP5 | TOP1 | TOP5 | TOP1 | TOP5 | TOP1 | TOP5 |
| *Abstract* | 40.65 | 63.92 | 47.38 | 75.17 | 53.87 | 81.69 | **61.75** | **89.11** | 61.07 | 88.24 |
| *Claims* | 39.37 | 65.42 | 48.09 | 77.78 | 56.10 | 83.40 | **63.40** | **90.22** | 62.82 | 89.46 |

**Table 5:** Performances of our models on the multi-class IPC classification of patent codes at the subclass level. Our DistilBERT models yielded the best performance overall under both abstract and claims input setups. TOP1 (i.e., accuracy) checks whether our prediction is the same as the actual label, whereas TOP5 measures whether the actual label of the input is amongst our five top predictions (i.e., five classes with the highest probability weights). Our high TOP5 scores indicate that our models are good at predicting the IPC codes of both popular and underrepresented classes. Figure 7 also provides evidence towards this conclusion.

trained on the claims section, achieves $63\%$ accuracy at TOP1 and $90\%$ accuracy at TOP5; this is notable since, as Figure 2 implies, a majority-class baseline could only yield around $10\%$ for TOP1 and $26.5\%$ for TOP5. In fact, the diagonal line in Figure 7 (center) also illustrates that the DistilBERT model, for instance, has learned features that enable it to successfully classify minority categories, such as F25C-*Producing, Working or Handling Ice* and D05B-*Sewing*. In general, the models trained on the abstract section performed as well as those trained on the claims section, perhaps an indication (in accord with the findings of Li et al. [3]) that the abstract alone contains useful information about the appropriate principal technology area the patent application might belong to. Model performance at the class level, [24] as opposed to the subclass level, was even stronger. The DistilBERT models achieve over $80\%$ TOP1 accuracy at the class level. Furthermore, incorrect predictions were often from similar or related technology areas. (Our high TOP5 scores also corroborate this finding.)

**Language Modeling.** We trained a masked language model on patent claims in the style of BERT [21]. We trained on a subset of the full patent dataset consisting of patent applications from 2011 to 2016 (1.57M applications) and evaluated on all patent applications from 2017 (187K applications). We performed masked language modeling with DistilRoBERTa [36] (82M parameters), initializing with a model pretrained on OpenWebText [37]. We release this model publicly so that researchers may utilize it for downstream tasks.[25]

**Abstractive Summarization.** We find that patent descriptions and claims can both be effectively summarized into patent abstracts, with claims leading to improved performance across all metrics. Qualitatively, the models can produce fluent abstracts complete with accurate details drawn from the claims section, as shown in Table 10. Consistent with BIGPATENT, our results suggest that the summarization of patent data could serve as a new domain-specific conditional generation task for the NLP community.

## 7 Evolution of Innovation Criteria and Trends over Time

A key advantage of HUPD is its combination of unstructured textual content and rich, structured bibliographic metadata. Patents, like many other applied natural-language domains, exhibit concept and domain shifts—the criteria for innovation varies across categories and evolves over time. In this section, we discuss some ways that our dataset exhibits time- and state-varying concepts. We hope this feature of patent data allows for fresh studies of the nature of shifts across categories and supports the development of NLP models that accommodate concept or distributional shifts.

**Universal Acceptance Classification.** We explored how a universal decision classifier, trained on all the IPC categories, might perform on individual categories. To this end, we trained a conditional DistilBERT classifier, where the conditional information had the title, issue year, and IPC code, in addition to the abstract section, to predict the acceptance likelihood of a patent application.[26] We used the patent applications filed between 2011 and 2016. The model achieved an overall accuracy score of $62\%$ on the entire class-balanced test set. When we assessed the model's performance on individual IPC subclasses, we discovered, for instance, that the model yielded almost $64.5\%$ accuracy on G06F-*Electric Digital Data Processing*, $61.9\%$ accuracy on H04N-*Pictorial Communication, e.g., Television*, and $57\%$ accuracy on G06Q-*Data Processing Systems or Methods*.

---

[24]We note that there are only nine broad IPC classes.

[25]We also use this model to visualize patent categories in Section H.

[26]The exact input setup of the conditional model was of the form: "[SEP] Title [Title] Year [Issue Year] IPC [IPC Subclass] Abstract [Abstract]"; the task was to predict the acceptance likelihood of a patent application.

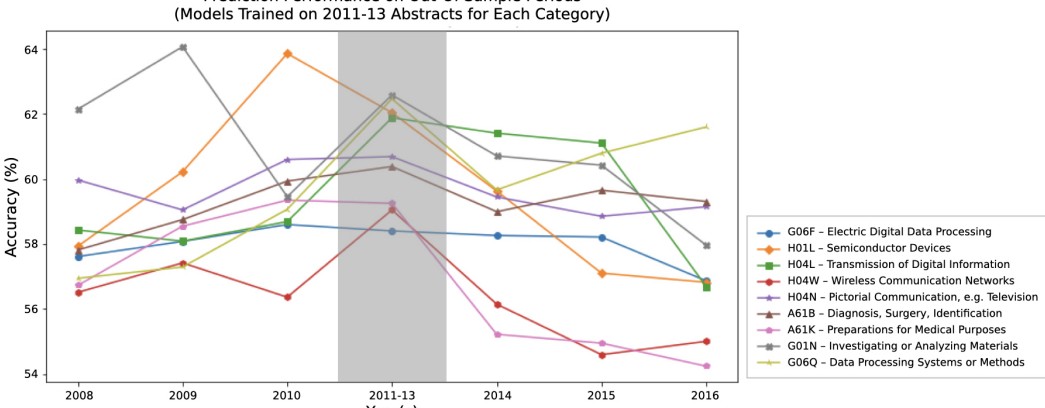

**Figure 3:** Performance of a BERT decision classifier trained on applications from 2011 to 2013 and evaluated on applications produced in earlier and later years. While model performance decays over time across most categories (suggesting changing acceptance standards), acceptance criteria appear to change more quickly in faster-moving fields (e.g., H01L-*Semiconductor Devices* and H04W-*Wireless Communication*) and slower in more developed fields (e.g., A61B-*Diagnosis, Surgery, Identification*). Of further interest may be the rise in turnover in acceptance criteria in G06Q-*Data Processing Systems or Methods* starting in 2014, which may coincide with the concurrent growth in development of B2B and fintech products. In future work, it may also be illustrative to use this dataset to investigate the impact of the U.S. Supreme Court decision in *Alice Corp. v. CLS Bank International*, 573 U.S. 208 (2014), on the software-related patent applications issued after 2014.

**Cross-Category Evaluation.** In order to understand and identify relationships between different patent evaluation criteria in different IPC classes, we also took each DistilBERT model trained on one IPC subclass of patents and evaluated it across all the other popular IPC subclasses. Figure 9 provides an illustration of our empirical findings. Patent technology areas that are conceptually closer to each other, such as G06F-*Electric Digital Data Processing* and H04L-*Transmission of Digital Information*, appear to have similar standards for patent acceptance. Notably, models trained in one context do not generalize to most other contexts. This suggests the criteria for patent acceptance are sensitive to the technical demands of the specific category of each application.

**Performance Over Time.** We can also use our models and dataset to understand the evolving criteria for patent acceptance, as well as innovation trends over time. Figure 3 shows the performance of a decision classification model trained on patent applications from 2011 to 2013 evaluated on applications produced in earlier and later years. While model performance usually deteriorates over time, suggesting changes in the features that predict patent acceptance, the rate of decay appears to be sharper for fields anecdotally thought to be faster-moving. This property of patent data may make it useful for studying concept shift that varies by class.

**Additional Tasks.** Given the richness of patent text and metadata, HUPD enables research on a wide range of tasks and use cases that may be explored in future work. In Section G, we describe some of these tasks—such as long sequence modeling, patent clustering, and patent examiner assignment—as well as the potential social impact and applications of our work.

## 8    Conclusion

We presented the Harvard USPTO Patent Dataset (HUPD), a large, versatile, and comprehensive structured corpus of patent data constructed for the NLP community. HUPD contains 4.5 million English-language utility patent applications filed to the USPTO between 2004 and 2018. We also established benchmarks for two classification-based and two generation-based tasks on our dataset. We provided detailed qualitative analyses of the models trained for these tasks and demonstrated how our dataset presents a setting with measurable concept shift. We hope that the combination of our dataset and the models used in this paper will not only advance research in patent analysis, but eventually also help patent applicants prepare more successful patent filings and provide a domain-specific laboratory for a multitude of NLP tasks.

## Acknowledgements

We thank Christopher Bavitz, Yonatan Belinkov, Tommy Bruzzese, Dallas Card, Jiafeng Chen, Kiran Dwivedi, Julia L. Englebert, Sebastian Gehrmann, Tayfun Gur, Umit Gurun, Dan Jurafsky, Peter Henderson, Ryan Kearns, Megan Ma, Daniel McFarland, Drew Pendergrass, Karen Sinclair, Kyle Swanson, Andrew A. Toole, Brandon Walton, Ian Wetherbee, Benjamin H. Wittenbrink, Michihiro Yasunaga, the members of the Lab for Economic Design at Harvard University, and attendees at our Stanford CodeX and Microsoft Research New England presentations for helpful comments and suggestions. We also would like to thank the anonymous reviewers for their valuable feedback and critiques. We are especially thankful to the USPTO for providing the bulk data and patent research datasets, and Andrew A. Toole, the Chief Economist at the USPTO, for his help navigating the USPTO's data products. We gratefully acknowledge the support of the three Microsoft Azure credit grants from the Harvard Data Science Initiative for data storage and computation. Some of the experiments presented in this paper were run on the FASRC Cannon cluster supported by the FAS Division of Science Research Computing Group at Harvard University and the Scalable Magic switch cluster. Suzgun gratefully acknowledges the support of a Harvard University Center of Mathematical Sciences and Applications (CMSA) Economic Design Fellowship and the Harvard College Research Program. Melas-Kyriazi gratefully acknowledges the support of a Rhodes Scholarship. Sarkar gratefully acknowledges the support of a National Science Foundation (NSF) Graduate Research Fellowship. Kominers gratefully acknowledges the support of the Ng Fund and the Mathematics in Economics Research Fund of the CMSA, as well as NSF grant SciSIP-1535813.

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

# A Data Card

## A.1 Dataset Description.

**Dataset Summary.** See Section 4.

**Languages.** The dataset contains English text only.

**Domain.** Patents (intellectual property).

**Additional Details.** The dataset contains utility patent applications filed to the USPTO between January 2004 and December 2018.

## A.2 Meta Information

**Dataset Curators.** The dataset was created by Mirac Suzgun, Luke Melas-Kyriazi, Suproteem K. Sarkar, Scott Duke Kominers, and Stuart M. Shieber.

**Licensing Information.** The dataset is released under the Creative Commons Attribution 4.0 International License.

**Leaderboard/Benchmarks.** The dataset has no associated public leaderboard; however, it introduces four useful benchmarks for the ML/NLP and Econ/IP communities, namely binary classification of patent decisions, multi-class IPC/CPC classification of patent codes at the subclass level, language modeling, and abstractive summarization. We release the models and code used in the main experiments on our GitHub codebase so that anyone can reproduce our empirical results, build their own models, make meaningful evaluations, and compare the performances of their models to ours.

For benchmarking purposes, we decided to split the data chronologically, establishing 2011-2016 as the training range and 2017 as the test range; however, this is not an absolute split for all the experiments. The users of our dataset can easily filter the metadata and fields, specify the years that they want to focus on, choose training and test sets as they wish, and thus create their own task or experiment setups. We believe the degree of freedom our dataset offers, as well as the allowance the dataset provides for different types of experiments and investigations that NLP and IP researchers and practitioners can take using the different data fields, is of paramount importance and novelty. In particular, we do not want, in any way, to restrict the usage and flexibility of the dataset by randomly or intentionally splitting according to some criteria (e.g., split the data chronologically, based on some certain grouping of IPC/CPC codes, etc.) for all the experiments.

## A.3 Dataset Structure

**Data Format and Structure.** Each patent application is defined by a distinct JSON file, named after its application number, and includes information about the application and publication numbers, title, decision status, filing and publication dates, primary and secondary classification codes, inventor(s), examiner, attorney, abstract, claims, background, summary, and full description of the proposed invention, among other fields. There are also supplementary variables, such as the small-entity indicator (which denotes whether the applicant is considered to be a small entity by the USPTO) and the foreign-filing indicator (which denotes whether the application was originally filed in a foreign country). In total, there are 34 data fields for each application. A full list of data fields used in the dataset is catalogued in the next section.

**Data Instances.** Each patent application in our patent dataset is defined by a distinct JSON file (e.g., 8914308.json), named after its unique application number. The format of the JSON files is as follows:

```json
1    {
2        "application_number": "...",
3        "publication_number": "...",
4        "title": "...",
5        "decision": "...",
6        "date_produced": "...",
7        "date_published": "...",
8        "main_cpc_label": "...",
9        "cpc_labels": ["...", "...", "..."],
10       "main_ipcr_label": "...",
11       "ipcr_labels": ["...", "...", "..."],
12       "patent_number": "...",
13       "filing_date": "...",
14       "patent_issue_date": "...",
15       "abandon_date": "...",
16       "uspc_class": "...",
17       "uspc_subclass": "...",
18       "examiner_id": "...",
19       "examiner_name_last": "...",
20       "examiner_name_first": "...",
21       "examiner_name_middle": "...",
22       "inventor_list": [
23           {
24               "inventor_name_last": "...",
25               "inventor_name_first": "...",
26               "inventor_city": "...",
27               "inventor_state": "...",
28               "inventor_country": "..."
29           }
30       ],
31       "abstract": "...",
32       "claims": "...",
33       "background": "...",
34       "summary": "...",
35       "full_description": "..."
36   }
```

**Data Fields.** In addition to distinct JSON files for each application, we consolidate patent metadata into a CSV file that links patent application numbers with both the metadata in the JSON files (e.g., *date_produced*, *main_cpc_label*) and additional features made available by the USPTO's PatEx data associated with the patent applications. These additional features are *atty_docket_number*, *file_location*, *wipo_pub_number*, *wipo_pub_date*, *patent_issue_date*, *small_entity_indicator*, *foreign*,*appl_status_desc*. We include code for querying subsets of patent applications by all the features we include for NLP experiments.

**Data Statistics.** The dataset contains 4,518,263 utility patent applications filed to the USPTO between January 2004 and December 2018.

## A.4 Dataset Creation.

**Source Data.** The Harvard USPTO Patent Dataset synthesizes multiple data sources from the USPTO: While the full patent application texts were obtained from the USPTO Bulk Data Storage System (Patent Application Data/XML Versions 4.0, 4.1, 4.2, 4.3, 4.4 ICE, as well as Version 1.5) as XML files, the bibliographic filing metadata were obtained from the USPTO Patent Examination Research Dataset (in February, 2021).

**Annotations.** Beyond our patent decision label, for which construction details are provided in Section 2, the dataset does not contain any human-written or computer-generated annotations beyond those produced by patent applicants or the USPTO.

**Personal and Sensitive Information.** The dataset contains information about the inventor(s) and examiner of each patent application. These details are, however, already in the public domain and available on the USPTO's Patent Application Information Retrieval (PAIR) system, as well as on Google Patents and PatentsView.

**Special Test Sets.** We presented four mainstream NLP experiments on HUPD, namely binary classification of patent decisions, multi-class IPC/CPC classification of patent codes at the subclass level, language modeling, and abstractive summarization. We reported our results in Section 6.

**Data Shift.** A major feature of HUPD is its structure, which allows it to demonstrate the evolution of concepts over time. As we illustrate in Section 7, the criteria for patent acceptance evolve over time at different rates, depending on category. We believe this is an important feature of the dataset, not only because of the social scientific questions it raises, but also because it facilitates research on models that can accommodate concept shift in a real-world setting.

**Sole Focus on the Textual Component.** Patent applicants often submit accompanying (black-and-white) drawings (i.e., graphs, charts, illustrations, diagrams and figures) in their applications to help the examiner and the reader to better understand their proposed invention. We initially considered including drawings in our dataset, in combination with all textual data, as we thought that having such a multimodal dataset would enable us to introduce new, domain-specific vision-and-language tasks into the field. However, we later came to learn that the Manual of Patent Examining Procedure states that a drawing in a patent application is required only if it is *totally necessary* for the understanding of the proposed invention (608.02 Drawing [R-10.2019]) (italicization was added by the authors). After making some preliminary estimations, we concluded that a multimodal patent dataset that contains both textual and visual components of the patent applications between 2004 and 2018 would be over 12 TB in size; therefore, it would be very difficult not only to host and store this amount of data and have it be easy to download and by users. In light of all these considerations, we thought that we should leave the incorporation of the visual material of patent applications for future work, and focused on the textual and bibliographic material of patent applications for the time being.

## A.5 Considerations for Using the Data

The dataset was created to build new and useful benchmarks for NLP and IP experiments, facilitate research on patent analysis, and eventually help small entities and businesses improve the quality of their patent applications with no cost.

**Social Impact of the Dataset.** We hope that our dataset will have a positive social impact on the ML/NLP and Econ/IP communities. We discuss these considerations in more detail in Section G.4.

**Impact on Underserved Communities and Discussion of Biases.** The dataset contains patent applications in English, a language with heavy attention from the NLP community. However, innovation is spread across many languages, cultures, and communities that are not reflected in this dataset. Our dataset is thus *not* representative of all kinds of innovation. Furthermore, patent applications require a fixed cost to draft and file and are not accessible to everyone. One goal of our dataset is to spur research that reduces the cost of drafting applications, potentially allowing for more people to seek intellectual property protection for their innovations.

**Limitations.** Please see the "Limitations" subsection of Section 4.

# B    Discussion of Potential Biases Embedded in the Dataset

In this section, we provide a further examination of our patent dataset for potential biases. For most of our analyses, we focus on the patent applications that were filed to the USPTO between 2011 and 2016 and investigate the correlation between inventor sex, entity size, and geographic location of patent inventors and patent outcomes. Our findings are consistent with the previous work [28–32], showing, among other things, that female inventors are notably underrepresented in the U.S. patenting system, that small and micro entities (e.g., independent inventors, small companies, non-profit organizations) are less likely to have positive outcomes in patent obtaining than large entities (e.g., companies with more than 500 employees), and that patent filing and acceptance rates are not uniformly distributed across the US. Our empirical findings suggest that any study focusing on the acceptance prediction task, especially if it is using the inventor information or the small-entity indicator as part of the input, should be aware of the the potential biases present in the dataset and interpret their results carefully in light of those biases.[27]

## B.1    Differences in Patent Filing: Inventor Sex and Gender Identity

The USPTO collects and provides a limited set of data fields linked to inventors and examiners—in particular, it specifies each inventor's name, along with the city, state, country of their residence, and primary examiner's name for each patent application. As noted by Motomura [38], research efforts that examine the link between demographic information (such as race, gender identity, sex, ethnicity, and socioeconomic class of inventors) and patent acceptance prospects need to match the USPTO's publicly available records with other sources. We cannot measure inventor sex or gender identity directly. However, we can estimate inventor sex from inventor names by using imputation procedures. Following the methodology of Jensen et al. [28], we used the data from the United States Social Security Administration (SSA), together with HUPD, to estimate the distribution of female and male inventors who filed patent applications to the USPTO.

The Social Security Administration data contains names obtained from Social Security card applications for births that occurred in the United States between 1880 and 2020 (inclusive) [39]. Each year contains an ordered list of tuples—where each tuple includes a name, assigned sex, and total number of Social Security card applications associated with that name and sex in that given year—all ranked by their popularity. Using this data, we could empirically estimate the assigned sex distributions for 100,364 unique names. For instance, under this model, the name "Julia" is a female name with $99.6\%$ probability, whereas the name "Taylor" is a female name with $74.4\%$ probability.

We took this basic statistical model to predict the assigned sex (male or female) of the inventors for each patent application filed between 2011 and 2016. To be consistent with the framework of Jensen et al. [28], we set a strict threshold and assumed that an inventor is female if this model predicts that the first name of the inventor is a female name with at least $95\%$ probability. Similarly, we assumed that an inventor is male if the model predicted the male class with at least $95\%$ probability.[28] Additionally, we labeled a name as unisex if it appeared in the SSA's collection of names but had a probability below the threshold value under the female and male categories, and as foreign-sounding[29] if it did not appear in the SSA's collection of names at all.[30] Though we had 2,149,443 patent applications, in total, for 2011-2016, we excluded 781,601 of these patent applications from our analyses, since they contained inventors all of whose names were labeled as foreign-sounding under the statistical model. At the end, the final sample pool included 1,367,842 patent applications that contained at least one identifiable name under the statistical model.

---

[27]These multifaceted biases are also likely to be present in models that are not making explicit usage of applicant, inventor, or other sensitive information due to correlated features, for instance.

[28]We acknowledge this labeling method is incomplete and makes many simplifying assumptions. See Jensen et al. [28] for further analysis.

[29]According to Masur and Ouellette [40], "[s]ince 2008, foreign inventors have filed for *more* U.S. patents each year than American inventors have." (p. 12)

[30]It is important to note that of all the 6,020,751 inventors who filed patent applications to the USPTO between 2011 and 2016, almost half of them had foreign-sounding names that could not be labeled as male, female, or unisex under this SSA-based model; most of these inventors were, as expected, inventors from foreign countries.

We found 87.5% of these patent applicants had at least one male inventor, whereas 17.2% of the same set of patent applications had at least one female inventor.[31] The numbers were similar for patent obtaining as well: Of the 554,380 granted patent applications containing at least one identifiable name under the statistical model, 88.0% of them had at least one male inventor, whereas 15.8% of them had at least one female inventor. These findings are consistent with the empirical findings of Jensen et al. [28] and reflect a sex disparity in both patent filing and patent obtaining in the United States. These sex differences in patent filings and obtaining indicate that female inventors are underrepresented in patenting, and may also have less success getting applications granted.

One family of analyses made possible by our dataset is a more systematic analysis of the disparities in patent acceptance probabilities by sex. For example, researchers could show that among patent applications with very similar distributions of text claims, sex predicts acceptance. Causal inference studies of this nature might provide more concrete tests for examiner bias in granting patent applications. But at the same time, the presence of sex disparities in the dataset may propagate biases, for example if inventor characteristics are included in the patent acceptance prediction task. We ask users of our dataset to consider these disparities carefully as they conduct their research.

### B.2  Differences in Patent Filing: Entity Size

The USPTO offers reduced application and maintenance fees for patent applicants who are qualified for the small or micro entity status, removing financial barriers to innovation for independent inventors and small businesses with limited resources. In our dataset, we include a small-entity indicator for each patent application—denoting whether the applicant is considered to be a small entity by the USPTO. Using HUPD, we explored how much the patent acceptance rates vary across different business sizes. As before, we restricted our focus to the 2011-2016 year range.

For clarity, we define three different entity statuses under the USPTO's application system. A small entity status is entitled to a patent applicant if the applicant is representing an individual (viz., independent inventor), representing a non-profit organization (e.g., university, 501(c)(3) organization, etc.), or a small business (e.g., a company that meets the standards set forth in 13 CFR 121.801 through 121.805). A micro entity is a small entity that meets additional criteria, such as not having been the inventor of a total of more than four patent applications. While small entities are eligible for a 50% discount, micro entities are eligible for a 75% discount. Applicants that pay regular application and maintenance fees (e.g., companies with more than 500 employees) are considered undiscounted (large) entities.

Figure 4 shows the distribution of patent outcomes across three entity categories—namely small, micro, and undiscounted (large)—between 2011 and 2016. Applications filed by small and micro entities constitute only a small fraction of the patent applications. Moreover, small entities have lower acceptance rates than large entities. Finally, large entities seem to not only file more CONT-applications but also have higher CONT-acceptance rates than small entities.

Additionally, we looked at the representation of female inventors in different entity categories using the assigned sex estimation model from the previous section. Of the patent applications that had at least one male, female, or unisex-sounding American name under the statistical model, 17.8% had at least one female inventor in their inventor list in the case of the small entities. This number was slightly higher (21.8%) for the micro entities and slightly lower (16.9%) for the large entities.

### B.3  Differences in Patent Filing: Geographic Location

The USPTO limits the inventor residence information to the city and state (or foreign country) in patent applications. In our examination, we measured the geographical distribution of US-based inventors at the state-level. The top-left plot in Figure 5 shows the geographical distribution of the state-level residences of the inventors who filed patent applications to the USPTO between 2011 and 2016. According to this plot, almost a quarter of the US-based inventors had a residence in California, the most populous U.S. state, at the time of their submissions. On the other hand, 574 inventors had a residence in Alaska. The top-right plot in Figure 5 shows the geographical distribution of the residences of the US-based inventors with granted patents (for the same year-range). Finally, the plot

---

[31]Even when we included the unisex category in our analysis and looked at the percentage of the patent applications that contained at least one female or unisex inventor name, the number only increased to 32.2%.

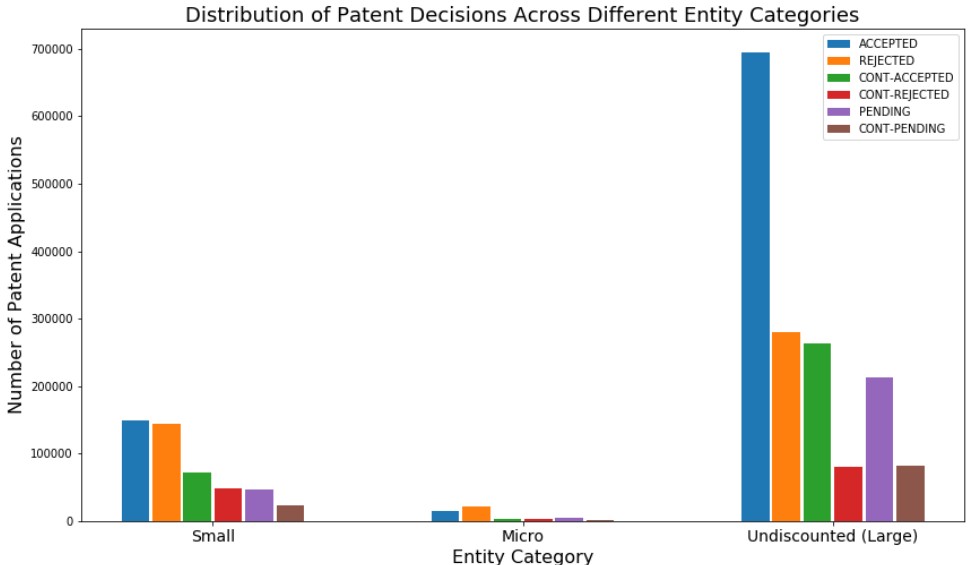

**Figure 4:** Distribution of patent decision outcomes across small, micro, and large entities from 2011 to 2016. Large entities have higher acceptance rates than small and micro entities.

in the second row illustrates the average patent acceptance rate of inventors from each state in the U.S. for the 2011-2016 year range. While the success rates vary across different parts of the country, some states with large corporations and concentrated entrepreneurship areas (including California, New York, Texas, and Michigan) appear to have noticeably high patent success rates. In fact, Michigan, with its $46.1\%$ success rate, has the highest patent acceptance rate amongst all the U.S. states, while Nevada ($23.1\%$) has the lowest. These results are consistent with the findings of Bell et al. [30].

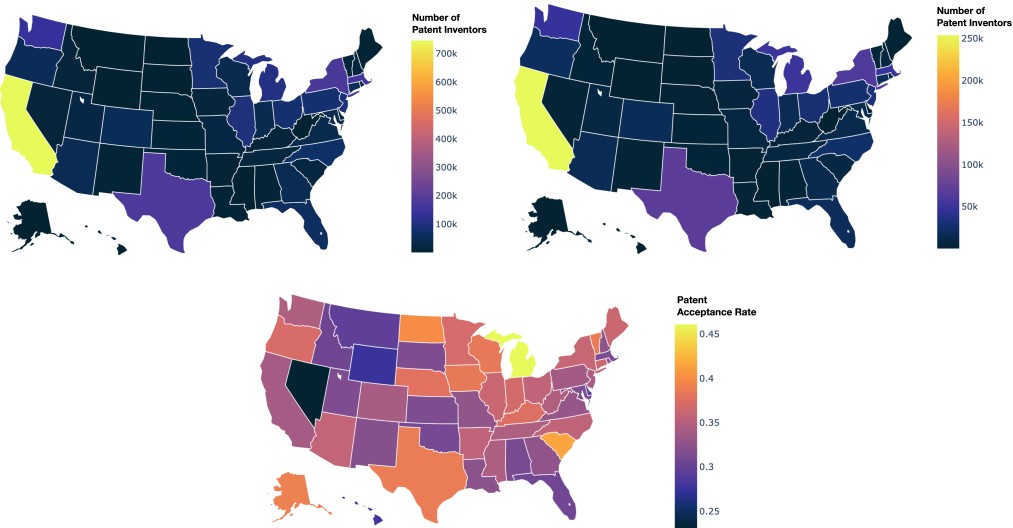

**Figure 5:** Top-left: The geographical distribution of the state-level residences of the US-based inventors who filed patent applications to the USPTO between 2011 and 2016. Top-right: The subset of the inventors from before who had granted patents. Bottom (second row): The patent acceptance rates across different states in the US.

## B.4 Differences in Patent Examination: Examiner Sex and Gender Identity

We additionally estimated the distribution of estimated assigned sex of the examiners at the USPTO. We used the same subset of the dataset as before and focused on the examiners of the patent

applications that were filed between 2011 and 2016. As shown on the left plot in Figure 6, of the 10,484 patent examiners in consideration, 58.8% of them had predicted male names, 20.6% had predicted female names, 11.4% had predicted foreign-sounding names, and the rest (9.2%) had unisex names, according to the basic statistical model used in Section B.1.

The right plot in Figure 6 illustrates the relationship between patent decision outcomes and estimated sexes of patent examiners. According to our estimation of assigned sex, and among examiners whose names we observe in the SSA data, male patent examiners have higher accepted rates than female patent examiners: The "Accepted"/"Rejected" ratio is close to 2.0 for male patent examiners and 1.4 for female patent examiners.

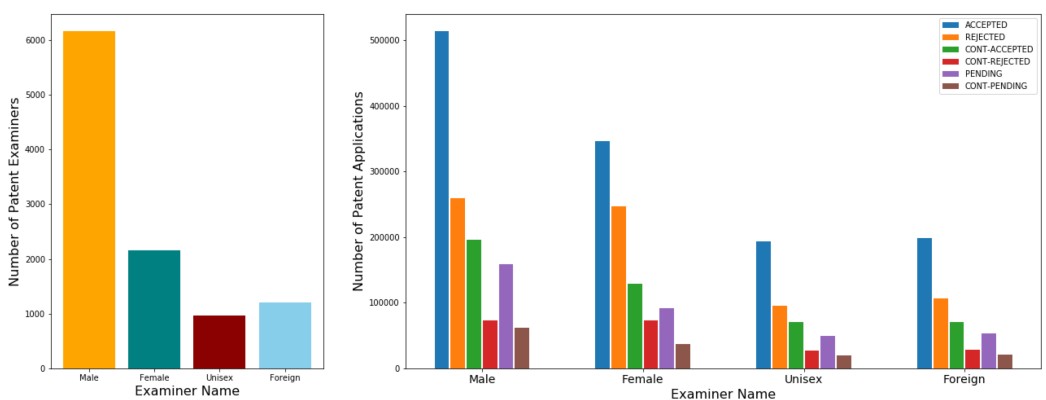

**Figure 6:** Relationship between patent decision outcomes and patent examiners' estimated sexes.

# C    Further Comparisons with Existing Datasets

## C.1    Comparison with BIGPATENT

It is natural to compare HUPD to one of the most widely-used existing patent datasets in NLP, BIGPATENT [4]. In comparison to BIGPATENT, HUPD contains not only significantly more patent documents (4.5 million vs. 1.3 million) but also much richer (bibliographic) meta-information about each patent document. Since BIGPATENT was proposed primarily for the task of abstractive summarization, it includes only four of the 34 data fields available in HUPD (publication number, application number, abstract, and description). Notably, BIGPATENT does not include the claims section, which is often considered to be the most important section for describing the contributions of an invention. Many tasks that can be performed with HUPD (analysis of text over time, fine-grained classification, etc.) are not possible to perform with the BIGPATENT data. Additionally, whereas BIGPATENT contains only granted patents, HUPD contains patent applications. As a result, prediction of patent acceptance, a new task we introduce to NLP, is not possible with BIGPATENT. Finally, the BIGPATENT is pre-tokenized using NLTK [41], which might cause issues for applications that contain chemical formulae and mathematical equations. HUPD, by contrast, provides the raw patent text and may be tokenized using a custom vocabulary for tasks in which it is important to correctly represent formulae or equations.

## C.2    Comparison with Large Scientific Text Corpora

One noteworthy domain-specific NLP corpus is Allen AI's Semantic Scholar Open Research Corpus (S2ORC; Lo et al. [9]), which contains 81 million English-language academic papers spanning multiple academic disciplines. Of these papers, the authors provide full structured text from 8.1M open-access PDF files and 1.5M LaTeX source files, along with their appropriate metadata. They include the title, author list, publication year, publication venue (or journal), abstract, paragraphs of the body of the text, section heading, figure and table information (along with their corresponding captions), equations, headers, footers, inline citations, and prior art as data fields in their JSONlines files. Due to the nature of academic papers, however, not all the papers have content for these data fields. With its size and comprehensive structured metadata, S2ORC is comparable to HUPD in its nature. Another concurrent work by Saier and Färber [42] introduced the unarXive dataset, a collection of over one million academic papers with links to almost 2.7 million unique publications; however, this dataset is not as comprehensive and well-structured as S2ORC or our dataset. Furthermore, Dasigi et al. [43] introduced Qasper, a dataset of information-seeking question-answering (QA) dataset over 1585 NLP papers, but their focus was limited to the evaluation of document-grounded QA models. Contract Understanding Atticus Dataset (CUAD; Hendrycks et al. [44]) is another annotated NLP dataset that contains more than 500 contracts with over 13,000 expert annotations and 41 label categories. CUAD is unique in the sense that the labels were annotated by legal experts and poses the identification of the relationship between different subtexts of a contact with different label categories as its primary task. WikiBio [10] is also a large scientific corpus that is worth mentioning: It contains almost 0.73 million unique biographies of famous people extracted from English Wikipedia, where each biography contains the first paragraph of the article and the infobox (fact table). The WikiBio dataset has been historically used for table-to-text generation [45], but it has been also adopted for question-answering [46]. We remark that our dataset provides a diverse range of language-based tasks to the community, including, but not limited to, binary classification of patent decision outcomes, multi-class IPC/CPC classification, patent clustering, long-sequence language modeling, abstractive summarization, document-level information extraction, named-entity recognition and extraction. HUPD is also notable in being one of the largest publicly-available collection of well-structured and domain-specific textual data. Finally, we remark that it has the potential to facilitate research and development in not only NLP but also IP.

## C.3    Comparison with Large-Scale Web-Scraped NLP Datasets

Given the size of HUPD (350GB of raw text), it is also possible to compare it to the extremely large-scale NLP corpora currently used for language model pretraining, such as Colossal Clean Crawled Corpus (C4) [47], the Pile [48], and OpenWebText [37]. These datasets contain vast quantities of text scraped from the Internet, some of which is derived from patents. In fact, a recent analysis of the C4 dataset [49] found that "patents.google.com" is the single most-frequent source of text in the corpus,

as measured by number of tokens. Furthermore, Dodge et al. [49] found that this patent text was not clean: A significant percentage was machine-translated from non-English languages and/or extracted from images with OCR. HUPD differs markedly from these web-scraped datasets because we obtain our text directly from the USPTO BDSS; textual content in our dataset is clean, carefully-extracted, and consistently-formatted. Additionally, whereas other large-scale text datasets typically contain only a stream of text, HUPD contains numerous structured data fields for each document.[32] The main focus of HUPD is not on pre-training large language models from scratch, but due to its large size and high quality, it can be a good complement to the existing extremely-large-scale web-scraped NLP corpora.

### C.4 Comparison with Popular Patent Search Tools and Large Repositories of Patent Data

In this section, we compare HUPD to popular English-language-based patent search tools and large repositories of patent data which are not constructed primarily for the ML/NLP community.

**Publicly Available Patent Search and Analysis Tools.** To the best of our knowledge, Google Patents is the most comprehensive and most diverse query-based patent search tool, containing over 120 million distilled patent documents (including patent applications, pre-grant publications, and granted patents), from more than one hundred patent offices all around the world. Similarly, PatentsView is a web-based patent data visualization and search tool that aims to improve the quality, impact, evaluation, and visibility of patents and patent applications filed to the USPTO. PatentsView is supported by the Office of the Chief Economist at the USPTO. The current PatentsView API allows each user to make up to 45 queries per minute. Using the API tool, users can get access to the full text of the title and abstract section of patent documents, as well the inventor, assignee, location, CPC/USPC classification details, among some other numerical and bibliographic statistics. The Lens Patent API allows search over 140 million patent meta-records worldwide. It has a free 14-day trial period (with limited API requests); afterwards, users need to pay fee for customized access, though pricing is determined on the nature and volume of the use case.

Aimed to replace the USPTO's existing search platform (Pub EAST, Pub WEST, Pat/FT, and App/FT) with a single, more comprehensive, and more modern online service, Patent Public Search is the USPTO's own recent search tool can perform and display searches over the USPTO's three big databases (namely, US-PGPUB, USPAT, and and USOCR). Patent Public Search is based on the Patents End-to-End (PE2E) search tool that the USPTO examiners use to identify prior art. There are additional online search tools developed and supported by different patent offices, such as WIPO Patentscope and Espacenet (supported by the WIPO and European Patent Office, respectively), that seek to improve public access to patent information.[33]

Managed by the EPO, the PATSTAT Global is a patent statistics database that contains bibliographic data and legal information about more than 110 million patent documents (patent applications and patents) from the patent offices of many industrial and developing countries from 1980 onwards.[34] While it has wide bibliographic coverage and is regularly updated, the PATSTAT database does require users to pay a subscription free for usage. It allows users to retrieve the title, abstract, inventor/application, classification code, technical field, other bibliographic metadata, and legal history of patent documents.[35]

**Large Repositories of Raw US Patent Data.** The USPTO makes several patent raw text datasets publicly available in XML formats.[36] As noted before, Patent Application Data/XML (Versions 4.0-4.4 ICE and Version 1.5) from the BDSS, together with the Patent Examination Research Dataset (PatEx) from USPTO's Public Patent Application Information Retrieval (PAIR) system, is the core

---

[32]Wikipedia contains structured fields alongside text for some documents. However, unlike patent applications and patent documents, the pages in Wikipedia contain *different* (non-uniform, varied) structured fields, making this information difficult to use across different tasks.

[33]Espacenet allows users to make queries in English, French, and German.

[34]The PATSTAT data is based on the EPO data and the data provided by other international patent offices on a voluntary basis. For more information: https://www.epo.org/searching-for-patents/business/patstat.html.

[35]We invite our readers to refer to the I³ Open Innovation Dataset Index (https://iiindex.org/) to learn more about innovation-related datasets, tools, platforms, and resources.

[36]The USPTO Bulk Data Storage System (BDSS) website: https://bulkdata.uspto.gov/.

data we have used to construct HUPD. The patent applications in the USPTO Bulk Data system are organized by their filing application years and concatenated in big bulk XML files. Each patent application needs to be extracted and unconcatenated back to individual XML documents for clean data processing purposes. Additionally, the USPTO makes text data and rich bibliographic information for patents available via its PatentsView service.

The Google Patents Public Datasets collection makes consistently-formatted patent text and filing information available to users, though its primary purpose is not to be used as an NLP dataset. The collection contains bibliographic information on more than 90 million patent documents from 17 countries, in addition to the full textual content of millions of US patent documents, provided by IFI CLAIMS Patent Services [50]. The data from this repository can downloaded using BigQuery.[37]

The MAtrixware REsearch Collection (MAREC) Dataset[38] is a static, standardized, and multilingual dataset of patent applications and granted patents. There are 19 million raw patent documents in the dataset—written in 19 languages (though the majority of them are written in English, German, and French), spanning an almost three-decade period (1976-2008), coming from the European Patent Office (EPO), Japan Patent Office (JPO), United States Patent and Trademark Office (USPTO), as well as the World Intellectual Property Organization (WIPO). The documents in the MAREC dataset are normalized to a uniform XML format. The standardized data fields are primarily the publication number, filing date, country of invention, language, citation information, inventor information, main subject classifications (e.g., IPC codes). Almost half of the documents contain full textual content of the inventions. The dataset, whose raw data size (almost 600GB) is comparable to ours, can be accessed via BigQuery and via bulk download.

We include a more directed comparison of HUPD with other patent datasets constructed for NLP patent analysis in Table 1 and Section C.1.

---

[37]The curators of the Google Patents Public Datasets also have an accompanying GitHub repository (https://github.com/google/patents-public-data) that explains how to use the dataset on BigQuery.

[38]MAREC's website: http://www.ifs.tuwien.ac.at/imp/marec.shtml.

# D Glossary

In this section, we provide a list of commonly used patent-related terms, concepts, and acronyms in this paper. While this list is not meant to be extensive or comprehensive, it is still inclusive and informative; it can be used by our readers as a reference guide as they read the paper. Unless otherwise marked with an asterisk * at the end, the definition for each term was taken from the USPTO's Glossary.[39]

- Abandonment: A patent application becomes abandoned for failure to file a complete and proper reply as the condition of the application may require within the time period provided under 37 CFR § 1.134 and § 1.136 unless an Office action indicates otherwise. Abandonment may be either of the invention or of an application. An abandoned application, in accordance with 37 CFR §§ 1.135 and 1.138, is one which is removed from the Office docket of pending applications.

- Applicant: Inventor or joint inventors who are applying for a patent on their own invention, or the person mentioned in 37 CFR 1.42, 1.43 or 1.47 who is applying for a patent in place of the inventor.

- Application filing date: The date the USPTO receives an application in English that includes all the following: (1) The applicant's name, (2) A name and address for correspondence (3) A clear drawing of the mark to be registered, (4) A list of the goods or services, and (5) An application filing fee for at least one class of goods or services.

- CIP (Continuation-in-Part): An application filed during the lifetime of an earlier nonprovisional application, repeating some substantial portion or all of the earlier nonprovisional application and adding matter not disclosed in the earlier nonprovisional application.

- Claims: define the invention and are what aspects are legally enforceable. The specification must conclude with a claim particularly pointing out and distinctly claiming the subject matter which the applicant regards as his invention or discovery. The claim or claims must conform to the invention as set forth in the remainder of the specification and the terms and phrases used in the claims must find clear support or antecedent basis in the description so that the meaning of the terms in the claims may be ascertainable (clearly understood ) by reference to the description. (See 37 CFR § 1.58(a)).

- Classification: Patents are classified (organized) in the U.S. by a system using a 3 digit class and a 3 digit subclass to describe every similar grouping of patent art. A single invention may be described by multiple classification codes.

- Continuation: A second application for the same invention claimed in a prior nonprovisional application and filed before the first application becomes abandoned or patented.

- Continuing application: A continuation, divisional, or continuation-in-part patent application.

- CPC: Cooperative Patent Classification.*

- Design patent: May be granted to anyone who invents a new, original, and ornamental design for an article of manufacture.

- Design patent application: An application for a patent to protect against the unauthorized use of new, original, and ornamental designs for articles of manufacture.

- Divisional application: A later application for an independent or distinct invention disclosing and claiming (only a portion of and) only subject matter disclosed in the earlier or parent application.

- Drawing (patent): Patent drawings must show every feature of the invention as specified in the claims. Omission of drawings may cause an application to be considered incomplete but are only required if drawings are necessary for the understanding of the subject matter sought to be patented.

- Express abandonment: A patent application may be expressly abandoned by filing a written declaration of abandonment identifying the application in the United States Patent and Trademark Office. Express abandonment becomes effective when an appropriate official

---

[39]USPTO Glossary: https://www.uspto.gov/learning-and-resources/glossary.

of the Office takes action thereon. Express abandonment of the application may not be recognized by the USPTO before the date of issue or publication unless it is actually received by appropriate officials in time to act. Abandonment may be either of the invention or of an application. An abandoned application, in accordance with 37 CFR 1.135 and 1.138, is one which is removed from the USPTO docket of pending applications.

- Filing date: the date of receipt in the Office of an application which includes (1) a specification containing a description and, if the application is a nonprovisional application, at least one claim, and (2) any required drawings.

- Final office action (rejection): An Office action on the second or any subsequent examination or consideration by an examiner that is intended to close the prosecution of a nonprovisional patent application. Applicant's reply under 37 CFR 1.113 to a final rejection is limited either to an appeal in the case of rejection of any claim to the Board of Patent Appeals and Interferences (37 CFR 1.191) or to an amendment complying with the requirements set forth in the Office action (37 CFR 1.114 or 1.116). Reply to a final rejection must comply with 37 CFR 1.114 or include cancellation of, or appeal from the rejection of, each rejected claim. If any claim stands allowed, the reply to a final rejection must comply with any requirements or objections as to form (37 CFR 1.113(c)).

- FY (fiscal year): The federal fiscal year extends from October 1 through September 30.

- Independent claim: A claim that does not refer back to or depend on another claim.

- Infringement (patent): Unauthorized making, using, offering to sell, selling or importing into the United States any patented invention.

- Inventor: One who contributes to the conception of an invention. The patent law of the United States of America requires that the applicant in a patent application must be the inventor.

- IP: Intellectual property.

- IPC: International Patent Classification.[*]

- Issue date: The date that a patent application becomes a U.S. patent. The issue date is the date that patent rights can be exercised. U.S. patents are always issued on Tuesdays.

- MPEP: Manual of Patent Examining Procedure.

- National stage application: An application which has entered the national phase of the Patent Cooperation Treaty by the fulfillment of certain requirements in a national Office, which is an authority entrusted with the granting of national or regional patents. Such an application is filed under 35 U.S.C. §371 in the United States and is referred to as a "371 application."

- Non-final Office action: An Office action letter that raises new issues and usually is the first phase of the examination process. An examining attorney will issue a non-final Office action after reviewing the application for the first time. If a new issue arises after the applicant responds to the first non-final Office action, the examining attorney will issue another non-final Office action that sets forth the new issue(s) and continues any that remain outstanding. Applicants must respond to non-final Office action letters within 6 months from the date they are issued to avoid abandonment of the application.

- Nonprovisional patent application: An application for patent filed under 35 U.S.C. 111(a) that includes all patent applications (i.e., utility, design, plant, and reissue) except provisional applications. The nonprovisional application establishes the filing date and initiates the examination process. A nonprovisional utility patent application must include a specification, including a claim or claims; drawings, when necessary; an oath or declaration; and the prescribed filing fee.

- Notice of abandonment: A written notification from the USPTO that an application has been declared abandoned or, in other words, is no longer pending. If the application was abandoned unintentionally or due to Office error, the applicant has a deadline of two months from the issue date of the notice of abandonment to file either (1) a petition to revive the application or (2) a request to reinstate the application.

- Notice of allowability: A notification to the patent applicant that the application has been placed in condition for allowance.

- Notice of allowance (NOA): A written notification from the USPTO that a specific mark has survived the opposition period following publication in the Official Gazette, and has consequently been allowed for registration. It does not mean that the mark has registered yet. Receiving a notice of allowance is another step on the way to registration. Notices of allowance are only issued for applications that have been filed based on "intent to use". The notice of allowance is important because the issue date of the Notice of Allowance establishes the due date for filing a statement of use. After receiving the Notice of Allowance, the applicant must file a statement of use or a request for an extension of time to file a statement of use within 6 months from the issue date of the notice. If thea pplicant fails to timely file a statement of use or a request for an extension of time to file a statement of use, the application will be abandoned.

- Office action: A letter from a trademark examining attorney setting forth the legal status of a trademark application. There are several types of Office actions: examiner's amendments, priority actions, non-final Office actions, final Office actions, and suspension inquiry letters.

- Parent application: The term "parent" is applied to an earlier application of the inventor disclosing a given invention.

- Patent: A property right granted by the Government of the United States of America to an inventor "to exclude others from making, using, offering for sale, or selling the invention throughout the United States or importing the invention into the United States" for a limited time in exchange for public disclosure of the invention when the patent is granted.

- Patent family: A patent family is the same invention disclosed by a common inventor(s) and patented in more than one country.

- Patent number: A unique number assigned to a patent application when it issues as a patent.

- PG Pub: Pre-Grant Publication of patent application at 18 months from priority date:

- Plant application (patent): Applications to protect invented or discovered, asexually reproduced plant varieties.

- Provisional patent application: A provisional application for patent is a U. S. national application for patent filed in the USPTO under 35 U.S.C. § 111(b). It allows filing without a formal patent claim, oath or declaration, or any information disclosure (prior art) statement. It provides the means to establish an early effective filing date in a nonprovisional patent application filed under 35 U.S.C § 111(a) and automatically becomes abandoned after one year. It also allows the term "Patent Pending" to be applied.

- PTO: Patent and Trademark Office, former designation for USPTO.

- Publication number: A number assigned to the publication of patent applications filed on or after November 29, 2000. It includes the year, followed by a seven digit number, followed by a kind code. Example 200011234567A1.

- USPTO: United States Patent and Trademark Office.

- USPC: United States Patent Classification.*

- Utility patent: May be granted to anyone who invents or discovers any new, useful, and nonobvious process, machine, article of manufacture, or composition of matter, or any new and useful improvement thereof.

- Utility patent application: Protect useful processes, machines, articles of manufacture, and compositions of matter.

- WIPO: World Intellectual Property Organization.

- Withdrawn patent: An allowed application for patent in which the applicant files correspondence to withdraw the patent from issue; thus preventing it from issuing on the patent issue date. The printed document is sometimes available on the day of publication, but is later retracted and will not be available in the patent database. No copy of the patent document will appear on the official USPTO web site.

# E  Additional Tables and Figures

## E.1  Binary Decision Classification

### E.1.1  Complete Version of Table 4

| IPC – *Section* | BernNB | MultiNB | Logistic | CNN | DistilBERT$^{FT}$ | BERT$^{FT}$ | RoBERTa$^{FT}$ |
|---|---|---|---|---|---|---|---|
| **GO6F** – *Abstract* | 61.86 | 61.47 | 58.24 | 60.97 | **61.53** | 61.28 | 61.31 |
| **GO6F** – *Claims* | **63.96** | 62.06 | 58.02 | 63.38 | 63.37 | 62.97 | 63.25 |
| **H01L** – *Abstract* | 58.98 | 59.05 | 58.54 | 60.71 | 61.46 | 61.85 | **61.85** |
| **H01L** – *Claims* | 60.97 | 60.29 | 59.53 | **62.63** | 62.50 | 61.61 | 61.94 |
| **H04L** – *Abstract* | 59.35 | 58.75 | 58.75 | 59.89 | **60.54** | 60.52 | 60.05 |
| **H04L** – *Claims* | 62.13 | 61.04 | 58.04 | **62.34** | 61.42 | 61.47 | 61.74 |
| **H04W** – *Abstract* | 56.01 | 55.20 | 55.79 | **57.82** | 56.42 | 56.39 | 57.01 |
| **H04W** – *Claims* | 57.76 | 56.85 | 55.58 | **59.72** | 58.97 | 58.94 | 59.22 |
| **H04N** – *Abstract* | 60.74 | 60.64 | 58.79 | 60.37 | **62.01** | 61.93 | 61.51 |
| **H04N** – *Claims* | 62.51 | 61.01 | 57.53 | **63.98** | 62.82 | 61.98 | 62.14 |
| **A61B** – *Abstract* | 59.15 | 58.81 | 57.31 | 58.75 | 58.36 | 59.58 | **59.66** |
| **A61B** – *Claims* | 59.30 | 59.12 | 57.25 | 59.49 | 60.15 | **61.20** | 61.00 |
| **A61K** – *Abstract* | 58.14 | 57.82 | 55.46 | 56.85 | **58.47** | 56.45 | 57.08 |
| **A61K** – *Claims* | 57.31 | 57.93 | 56.20 | **59.06** | 58.72 | 57.91 | 57.84 |
| **GO1N** – *Abstract* | 59.85 | 59.89 | 57.25 | 59.98 | 59.00 | 60.30 | **61.10** |
| **GO1N** – *Claims* | 58.06 | 57.97 | 58.37 | 59.80 | 60.16 | 60.34 | **60.97** |
| **GO6Q** – *Abstract* | 61.53 | **61.64** | 58.52 | 60.46 | 61.23 | 61.09 | 61.56 |
| **GO6Q** – *Claims* | **63.96** | 63.31 | 57.17 | 62.90 | 61.88 | 62.19 | 63.25 |

**Table 6:** (Complete version of Table 4) Baseline performances of our models on the binary classification of patent decision task. All the models were trained and evaluated on the patent applications filed to the USPTO between 2011 and 2016. We note that BernNB and MultiNM denote Bernoulli and Multionomial NB classifiers trained on world-level unigrams (with minimum frequency of 3), respectively; Logistic a logistic regression model consisting of an embedding layer followed by a single linear layer trained on world-level unigrams; and CNN a Convulational Neural Network with a 2-D convolutional layer and a max-pooling layer trained on world-level unigrams (with minimum frequency of 3). The superscript $^{FT}$ on the Transformer models denotes that these models were fine-tuned, not trained from scratch.

### E.1.2  Top IPC/CPC Codes and Their Brief Descriptions

| IPC/CPC Code | Description |
|---|---|
| **G06F** | Electric Digital Data Processing |
| **H01L** | Semiconductor Devices |
| **A61K** | Preparations for Medical Purposes |
| **H04L** | Transmission of Digital Information |
| **H04N** | Pictorial Communication, e.g. Television |
| **A61B** | Diagnosis, Surgery, Identification |
| **G06Q** | Data Processing Systems or Methods |
| **H04W** | Wireless Communication Networks |
| **G01N** | Investigating or Analyzing Materials |

**Table 7:** Brief descriptions of the most common nine IPC/CPC subclass codes present in our year ranges.

## E.2 Multi-Class IPC/CPC Classification

### E.2.1 Words with the Highest Weights in the Bernoulli NB Classifier

| IPC | Words with Highest Weights |
|-----|----------------------------|
| G06F | data, includes, device, semiconductor, system, substrate, one, method, second, least, structure, display, information, plurality, layer, present |
| H01L | user, film, second, access, data, one, semiconductor, wherein, includes, memory, unit, region, device, portion, operation, area, used, layer |
| A61K | device, including, computer, least, image, forming, layer, files, data, part, package, driving, conductive, includes, processing, direction |
| H04L | data, includes, semiconductor, device, disposed, system, one, substrate, method, layer, display, second, plurality, least, structure, content |
| H04N | provided, data, includes, one, device, semiconductor, method, formed, second, system, electrical, least, substrate, display, information, electrode |
| A61B | data, one, includes, device, method, substrate, electrode, second, semiconductor, memory, least, light, unit, layer, application, set, plurality |
| G06Q | data, device, substrate, includes, semiconductor, one, method, structure, system, display, present, plurality, image, non, least, n, content |
| H04W | semiconductor, data, device, layer, includes, disposed, high, one, display, substrate, method, second, structure, system, machine, plurality |
| G01N | data, device, includes, one, semiconductor, method, bus, least, layer, including, memory, substrate, image, computer, content, second, application |

**Table 8:** Examples of words (excluding stopwords) with highest probability weights in their respective IPC subclasses in our Bernoulli naive Bayes classifier trained to predict the IPC code of a patent application based on the words in its abstract section. For a given IPC subclass label $y$, we looked at the probability value $p(x_i|y)$ for each word $x_i$ in the vocabulary, and listed the words with the highest probability values.

### E.2.2 Confusion Matrix

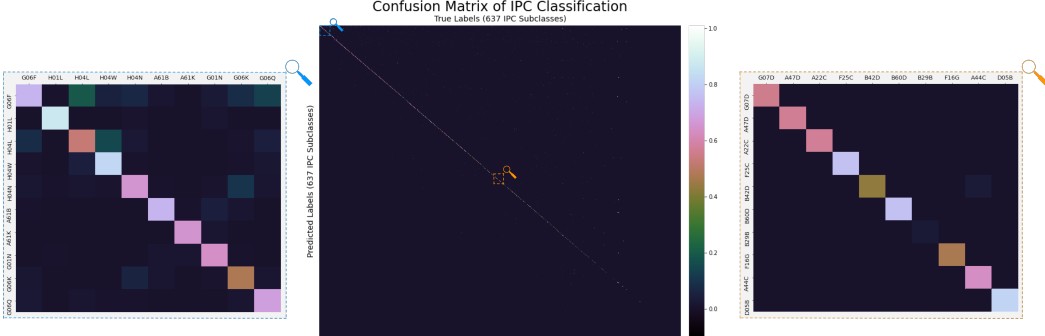

**Figure 7:** Confusion matrix of IPC code classification at the subclass level. This matrix was obtained from the DistilBERT model that was fine-tuned on the abstract sections. The IPC codes were ordered by their sizes from left to right and from top to bottom, respectively. The light diagonal line present on the center figure represents high recall values. The diagonal line disappears towards the lower right corner, since patents belonging to those IPC subclasses do not appear in our test set.

### E.2.3 Saliency Maps

To have a better understanding and appreciation of the behavior of our neural classifiers, we also made queries to identify which parts of the input texts our models might be attending to when making their predictions. To that end, we availed ourselves of various simple gradient-based saliency methods, such as SmoothGrad [51] and Integrated Gradient (IG; [52]).[40] We once again turned to the DistilBERT models for our analyses due to their relatively better performance and their potential to learn more complex representations. Figure 8 illustrates how much each input feature contributes to the prediction made by the DistilBERT model (trained on the abstract section as shown in Table 5). This analysis suggests the tokens that the model is paying attention to tend to be those that are most indicative of the technology area that each patent belongs to.[41]

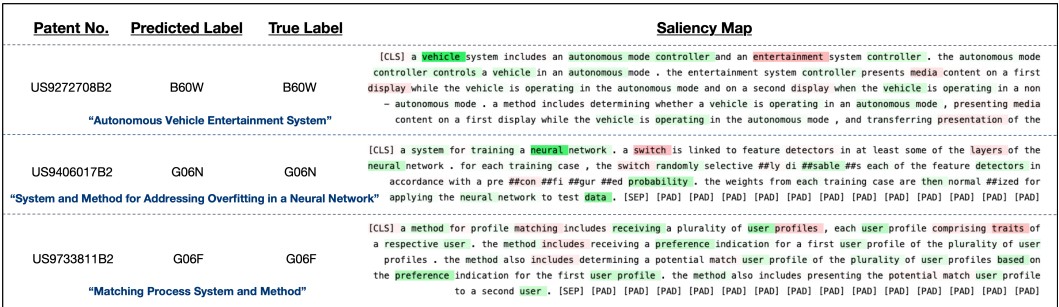

**Figure 8:** Saliency maps generated using Integrated Gradient [52] for the DistilBERT model trained to predict the IPC subclass of a patent application based on its abstract section. All our visualizations were created using the Captum library [53]. The green color highlights tokens that contribute more heavily toward the class label, and the red color signifies tokens that contribute less strongly toward the class label. Our models predicted the class label correctly in each example. These three examples are the abstract sections of three patent applications by Ford, Google, and Tinder, respectively. The input tokens that contribute the most to our predictions appear to be associated with the primary technology areas of these patent applications. These qualitative findings confirm our belief that our models have learned domain-specific features about IPC subclasses.

---

[40]We used the PyTorch-compatible interpretability library, Captum [53], in our experiments.

[41]On the other hand, the saliency maps of the DistilBERT models trained on both the abstract and the claims for the binary classification of the patent decision task on a couple of examples were less insightful.

## E.3 Abstractive Summarization

| Model | Setting | R-1 | R-2 | R-L |
|---|---|---|---|---|
| T5 (Small) | Description→Abstract | 62.87 | 47.20 | 54.36 |
| T5 (Small) | Claims→Abstract | 69.00 | 53.82 | 59.88 |

**Table 9:** Performances of our T5 summarization models on HUPD, as measured by ROUGE. Higher numbers reflect better performance. Summarization from the claims performs better than summarization from the description. Although our ROUGE scores are higher than those reported by Sharma et al. [4] on BIGPATENT, the scores are not directly comparable, since the evaluation data and tokenization schemes are different.

### E.3.1 Generated Summaries

| | |
|---|---|
| **Claims** | 1. An optical coherent receiver for an optical communication network, said optical coherent receiver being configured to receive a modulated optical signal and to process said modulated optical signal for generating an in-phase component and a quadrature component, said in-phase component and said quadrature component being electrical signals, said optical coherent receiver comprising a power adjuster in turn comprising: a multiplying unit configured to multiply said in-phase component by an in-phase gain thereby providing a power-adjusted in-phase component, and to multiply said quadrature component by a quadrature gain thereby providing a power-adjusted quadrature component; and a digital circuit connected between output and input of said multiplying unit and configured to compute: a common gain indicative of a sum of a power of said power-adjusted in-phase component and a power of said power-adjusted quadrature component, and a differential gain indicative of a difference between said power of said power-adjusted in-phase component and said power of said power-adjusted quadrature component; and said in-phase gain as a product between said common gain and said differential gain, and said quadrature gain as a ratio between said common gain and said differential gain. 2. An optical coherent receiver according to claim 1, wherein it further comprises an analog-to-digital unit connected at the input of said power adjuster, said analog-to-digital unit being configured to ... |
| **Generated Abstract** | An optical coherent receiver for an optical communication network is provided. The optical coherent receiver is configured to receive a modulated optical signal and to process the modulated optical signal for generating an in-phase component and a quadrature component. The in-phase component and the quadrature component are electrical signals. The optical coherent receiver includes a power adjuster in turn including a multiplying unit and a digital circuit. The multiplying unit is configured to multiply the in-phase component by an in-phase gain thereby providing a power-adjusted in-phase component, and to |
| **Ground-Truth Abstract** | It is disclosed an optical coherent receiver for an optical communication network. The optical coherent receiver is configured to receive a modulated optical signal and to process it for generating an in-phase component and a quadrature component. The optical coherent receiver comprises a power adjuster in turn comprising a multiplying unit and a retroactively connected digital circuit. The multiplying unit is configured to multiply the in-phase and quadrature components by in-phase and quadrature gains, respectively, thereby providing power-adjusted in-phase and quadrature components. The digital circuit is configured to compute: a common gain indicative of a sum of the powers of the power-adjusted in-phase and quadrature components; a differential gain indicative of a difference between the powers of the power-adjusted in-phase and quadrature components; and the in-phase and quadrature gains as a product and a ratio, respectively, between the common gain and the differential gain. |
| **Claims** | 1. A method of assessing the arterial health of an individual comprising: a) positioning one arm of the patient such that the arm is at approximately the same height as the heart of the individual; b) attaching a fingertip photoplethysmography device to a fingertip of the hand of said arm of the individual; c) measuring the analog pulse contour of the individual using the fingertip photoplethysmography device; d) digitizing the analog pulse contour; e) analyzing the digitized analog pulse contour for stable waveforms; f) processing the stable waveforms of the digitized analog pulse contour using dynamic time warping; g) comparing the stable waveforms to a library of known disease state waveforms; and h) assigning a most probable disease state for the individual based on said comparison. 2. The method according to claim 1 wherein the stable waveforms in step (e) are identified by analyzing the digitized pulse contour for two consecutive pulse heights that are within 10% of the same value and the next five waveforms are measured and stored as the stable waveforms. 3. The method according to claim 1 wherein following step (e), the stiffness index is calculated. 4. The method according to claim 1 wherein following step (e), the reflection index is calculated. 5. A method of generating a library of waveforms from individuals having a known arterial disease comprising: a) positioning one arm of an individual having a known arterial disease such that the arm is at approximately the same height as the heart of the individual; b) attaching a fingertip photoplethysmography device to a ... |
| **Generated Abstract** | A method of assessing the arterial health of an individual includes positioning one arm of the patient such that the arm is at approximately the same height as the heart of the individual; attaching a fingertip photoplethysmography device to a fingertip of the hand of said arm of the individual; measuring the analog pulse contour of the individual using the fingertip photoplethysmography device; digitizing the analog pulse contour; analyzing the digitized analog pulse contour for stable waveforms; processing the stable waveforms of the digitized analog pulse contour using dynamic time warping; |
| **Ground-Truth Abstract** | The arterial health of an individual can be determined by: attaching a fingertip photopiethysmography device to a fingertip of the hand of the elevated arm of the individual; measuring the analog pulse contour of the individual using the fingertip photopiethysmography device; digitizing the analog pulse contour; analyzing the digitized pulse contour for stable waveforms; processing the stable waveforms of the digitized pulse contour using dynamic time warping; comparing the stable waveforms to a library of known disease state waveforms; and assigning a most probable disease state for the individual based on said comparison. |

**Table 10:** Examples of claims summaries produced by our T5-Small model. Qualitatively, the models produce fluent patent abstracts complete with accurate details drawn from the claims section. Note also how different the structure of the above legal language is from most text used to train large language model; often, the entire abstract is a single long and complex sentence.

## E.4 Cross-Category Evaluation of BERT Acceptance Prediction

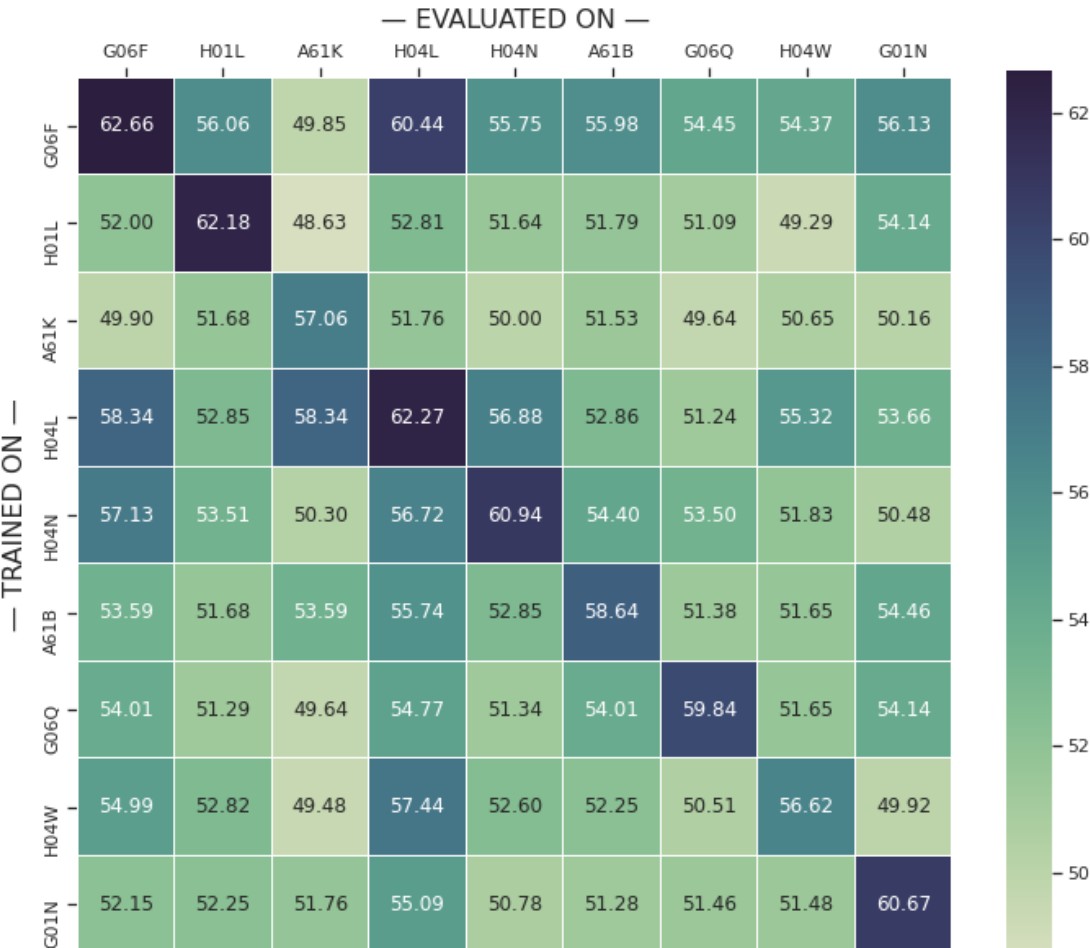

**Figure 9:** Cross-category evaluation of BERT acceptance prediction classifiers trained on one IPC code evaluated to predict acceptance on other IPC codes. Patent categories that are conceptually similar appear to have closer shared criteria for acceptance (for example, G06F-*Electric Digital Data Processing* and H04N-*Pictorial Communication*). Most models also tend to perform well when evaluated on patent applications in H04L-*Transmission of Digital Information*; moreover, the model trained on H04L-*Transmission of Digital Information* has high predictive power across other application types. This might reflect acceptance criteria in this category involving more high-level evaluations of quality, as opposed to more domain-specific innovations in engineering or manufacturing.

# F Experimental Details

## F.1 Binary Decision Classification

For this first task, we looked at the patent applications that were filed to the USPTO between January 2011 and December 2016 and excluded the pending applications from our experiments. Initially, we trained individual domain-specific classifiers, ranging from NB classifiers to RoBERTa, to predict the acceptability of patent applications in the most common IPC subclasses (see Figure 2). We used both the abstract and the claims, though separately.

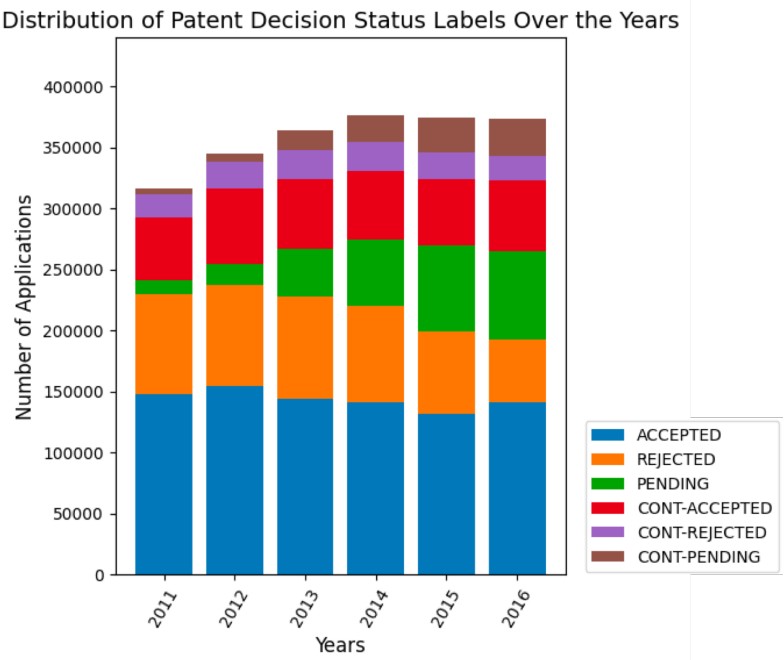

**Figure 10:** Distribution of decision status labels for patents filed between 2011 and 2016. Note that the relative share of pending to rejected applications increases over time as certain patents remain under review beyond the end of our metadata collection period. Approximately three quarters of patent applications are labeled as new filings.

To address the issue of imbalance in our decision status labels (see Figure 10), we used a weighted random sampler to select samples. We then randomly apportioned the data into training and test sets with an 85-15 split, but fixed the random seed across each model in each category to ensure fair comparability.

Our baselines for this task consisted of various subsets of Bernoulli and Multinomial naive Bayes classifiers, logistic regression (Logistic), CNN, DistilBERT [36], BERT [21], DistilROBERTa [36], RoBERTa [23], and T5-Small [54] for different tasks.[42]

## F.2 Abstractive Summarization

We converted the summarization task into a language modeling task using the setup described by Raffel et al. [54]. The source text and summary were simply concatenated, separated by a separation phrase ("summarize:"). We used the T5-Small "Text-to-Text Transformer" architecture of Raffel et al. [54] with 60 million parameters. We initialized with a model pretrained on the C4 dataset (`t5-small` on HuggingFace Transformers), although we found that training from scratch produced nearly the same performance due to the size of our dataset.

---

[42]We used scikit-learn [55] to train and evaluate our NB classifiers and HuggingFace's Transformers codebase [56] to implement and finetune our Transformers. We used HuggingFace's Tokenizer library for pre-processing and tokenization: For NB classifiers, Logistics, and CNNs, we adopted the "WordLevel" tokenizer; for each Transformer model, we automatically selected and applied its own associated tokenizer.

We used the same training-validation subsets used in the language modeling task: Applications from 2011-2016 for training and those from 2017 for validation. We truncated the training source texts (i.e., the claims/description) to a maximum of 1024 tokens. We trained for three epochs using the Adam [57] optimizer with batch size of 32 and a learning rate of $5 \cdot 10^{-5}$.

### F.3 Special Focus on the 2011-2016 Year Range in Our Experiments

While the dataset consists of the patent applications filed to the USPTO between 2004 and 2018, we primarily focused on the 2011-2016 year range in our main NLP experiments. We acknowledge that the primary reason behind this specification was the limited set of compute resources we had to train and deploy our models: We had limited GPU credits and memory bandwidth at our disposal when we were training and fine-tuning our models. We ultimately decided to focus on the 2011-2016 range, because the patent applications from this time period (i) seem to reflect recent technical developments and innovations, (ii) cover a good variety of different industries and technology areas, and (iii) could be easily used to train classification, language, and summarization models under one day to study the characteristics and evolution of patent-driven innovation in the United States. Furthermore, we used the same subset throughout most of our experiments for consistency and coherency—and this also allowed us to easily cache our data and get access to it almost instantly, rather than pre-processing, tokenizing, and batching each time for a new year range. We demonstrate the utility and benefits of the dataset on an exemplary subset, but we wish to remind our readers that a major goal of this dataset is to facilitate future research that can use the comprehensive time variation and rich metadata that we make available. Therefore, we hope that these considerations justify the special focus on the patent applications in the 2011-2016 year range.

# G Discussion of Additional Tasks

## G.1 Long Sequence Modeling

One interesting application of HUPD that we have not yet explored is to use its patent description field as part of a benchmark for long-sequence modeling. Long-sequence modeling has recently gained significant attention due to the proliferation of efficient Transformer architectures (e.g., Longformer, Performer). Popular benchmarks for these models include synthetic tasks, byte-level text tasks, and computer vision tasks; there does not exist a large-scale long-sequence word-level NLP benchmark since it is difficult to collect a large corpus of narrowly-focused long text documents. The task of language modeling of patent descriptions might be well-suited for this purpose; HUPD contains over 4.5M descriptions with an average length of 11,855 tokens (see Table 2), which far exceeds the context length of traditional Transformer models.

## G.2 Patent Clustering

A second application of the patent data is patent clustering: Given a patent application, one is tasked with finding similar patents (or patent applications). This tasks is particularly interesting to the IP community, since patent lawyers and examiners are interested in finding prior art for a given invention. To define clusters and find related patents, it would be possible to use a combination of metadata fields including fine-grained IPC/CPC classification codes, publication date, and inventor information.

## G.3 Patent Examiner Assignment

A third application of the patent data makes use of the examiner field, which has not been used in the tasks discussed above. This task involves predicting the examiner to which a new patent application should be assigned. This task may be viewed as an even more-fine-grained version of patent classification, because patent examiners often focus on a very narrow sub-field in which they have expertise or knowledge. We note that this task is possible under our framework because HUPD contains both raw texts of patent applications and also rich bibliographic metadata about each patent application.[43]

## G.4 Potential Social Impact and Applications & Future Work

In recent years, the USPTO has introduced a pilot program, called "Pro Se Assistance Program", to help small businesses and independent inventors file patent applications without enlisting the aid of a registered patent attorney or a legal agent. This initiative aims to enhance the quality of applications without putting any further financial burden on the shoulders of patent applicants who might have limited resources and means, as well as educating inventors about intellectual property protection and the patent filing process [59]. We hope our patent dataset and models might also provide some assistance to independent inventors and small businesses in improving the overall textual quality of their patent applications, to classify the technology areas of their inventions accurately, and to generate the abstract sections of their applications from their patent claims and/or description.

Given the scope of this paper, we could not conduct experiments on patent clustering, patent value prediction, or identification of "superstar" inventions. However, we foresee these research directions as direct valuable and impactful applications of our dataset, and suggest the research community push the boundaries on these fronts. In addition, our dataset's combination of natural language with structured metadata makes it an ideal gymnasium for evaluating and developing models that address concept shift across contexts and over time.

---

[43]It is of vital importance to assign a new patent application to an examiner who is skilled and qualified to understand the merits and shortcomings of the application in a critical and timely manner. And this is not a problem unique to patents: Shah [58], for instance, discusses the challenges and potential solutions in the current peer-review systems in scientific research, highlighting the importance of the assignment of the reviewers to papers and explaining the detrimental effects of the misutilization and mismatching of reviewer expertise. Shah [58] proposes using a two-stage assignment procedure, wherein the similarity score for every paper-reviewer pair is computed first and the maximization of the overall "utility" of paper-reviewer assignment is performed using the computer similarity scores second. Such an assignment procedure based on sum-similarity optimization methods can be studied using our dataset, for instance.

# H  Patent Category Visualizations

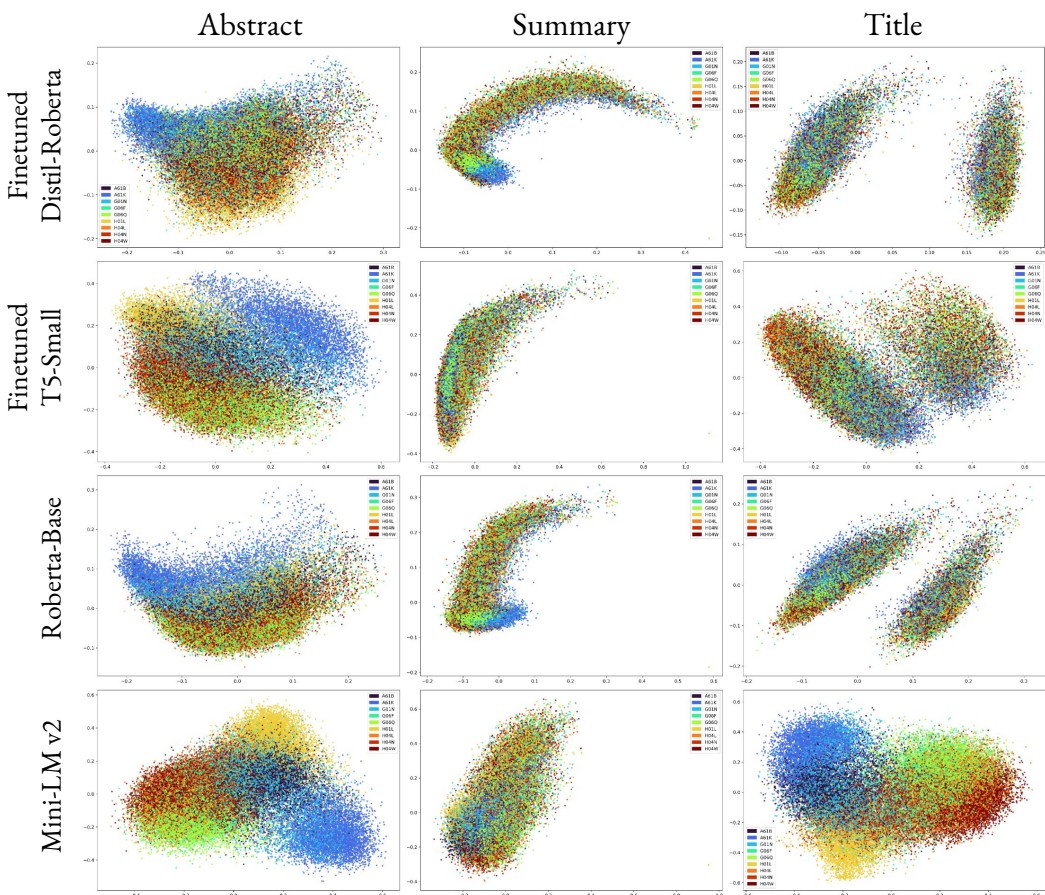

**Figure 11:** Visualizations of the vector representations of the data fields (abstract, summary, and title) of patent applications, embedded using four different Transformer models. Of these four models, two were fine-tuned on HUPD (Fine-tuned DistilRoBERTa and Fine-tuned T5-Small) and two were off-the-shelf, pre-trained models (RoBERTa-Base and Mini-LM v2). The Mini-LM v2 model was trained on a paraphrase dataset to produce good sentence-embeddings for text clustering. To create the figures above, we randomly sampled 500 patent applications from each of the top nine IPC categories for each year from 2008 to 2018. For each data field and each model, we computed embedding vectors using the model and reduced the dimensionality of these vectors using PCA. The above plots show the results of this dimensionality reduction, colored by IPC codes. We see that categories cluster strongly and similar categories (e.g., H04W-*Wireless Communication* and H04L-*Transmission of Digital Information*) are often close in embedding space.

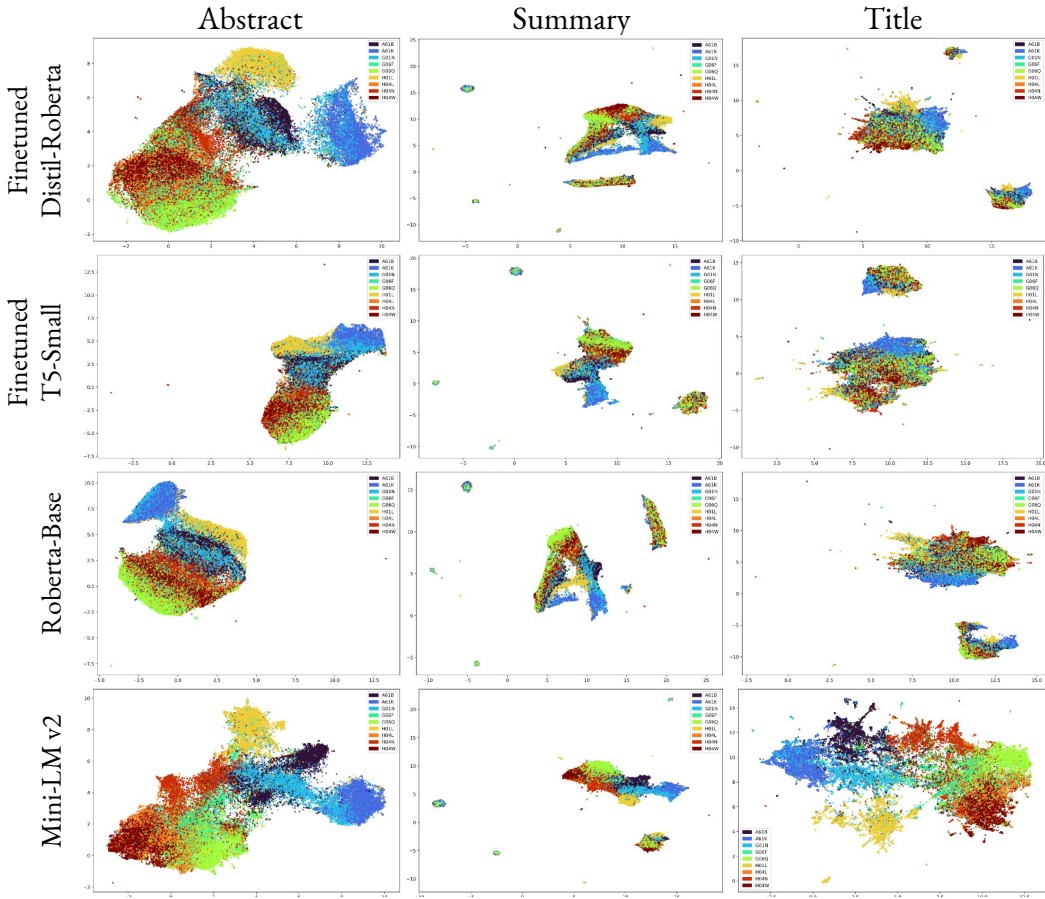

**Figure 12:** Visualizations of the vector representations of the data fields (abstract, summary, and title) of patent applications, embedded using four different Transformer models. This figure is similar to Figure 11, but uses UMAP [60] for dimensionality reduction rather than PCA.

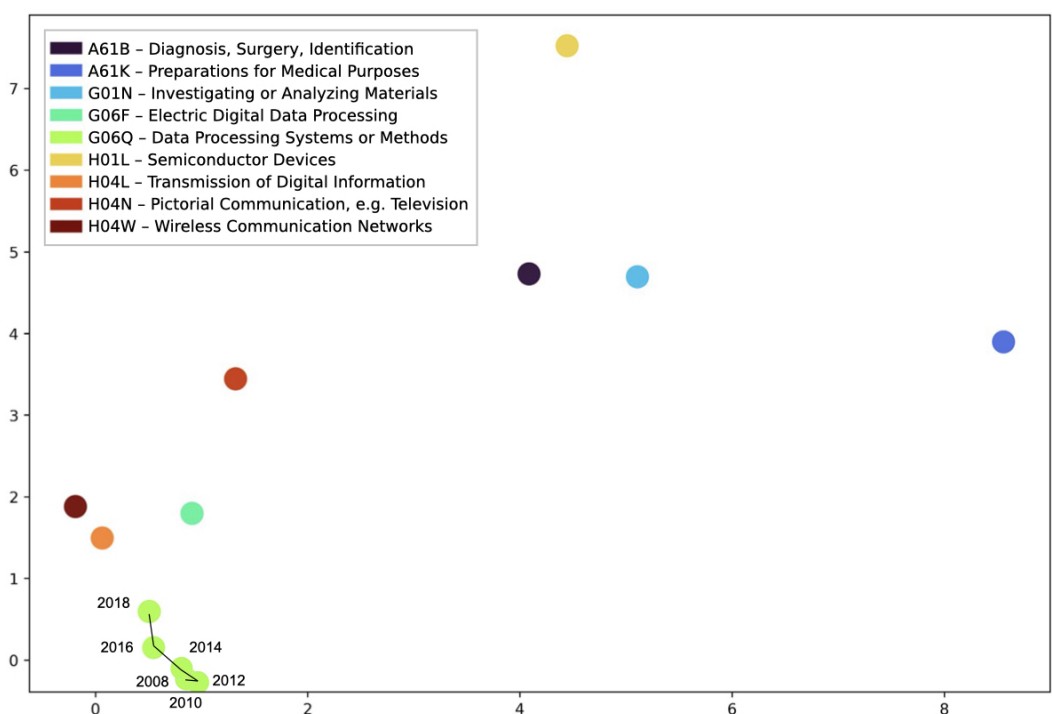

**Figure 13:** Depiction of the evolution over time of the averaged embeddings of the abstracts of patent applications from G06Q-*Data Processing Systems or Methods* relative to the other popular IPC codes in our dataset. To create this figure, as in the figures above, we randomly sampled 500 patent applications from each of the top nine IPC categories for each year from 2008 to 2018. Using our custom DistilRoBERTa model, we computed embedding vectors for the abstracts of each patent application and then reduced the dimensionality of these vectors using UMAP. The figure above shows the average UMAP embedding of the G06Q category for every other year, along with the average embeddings of the other categories (averaged across all samples from all years). The movement of the centroids of the G06Q category might be consistent with covariate shift over time in the distribution of language of patents in the category. G06F-*Electric Digital Data Processing* seems to move closer to H04L-*Transmission of Digital Information*/H04W-*Wireless Communication*. This evolution seems to be consistent with the digitization of data processing methods over the past two decades.

