# OpenReview forum: "The Harvard USPTO Patent Dataset: A Large-Scale, Well-Structured, and Multi-Purpose Corpus of Patent Applications"
_NeurIPS.cc/2023/Track/Datasets_and_Benchmarks — NeurIPS 2023 Datasets and Benchmarks Spotlight_

### Official Review · Reviewer_sQe3 · 2023-07-05
**A Large-Scale, Well-Structured, and Multi-Purpose Corpus of Patent Applications**

**Rating:** 7
**Confidence:** 4
**Correctness:** yes
**Clarity:** Yes, the paper is well written.

**Strengths:**

1. This dataset is large, well-structured, with complete information and a long time span.
2. Unlike the previous dataset, its a pre-application patent.


**Additional Feedback:**

See above

**Documentation:**

yes

**Ethics:**

yes

**Limitations:**

It will be more interesting to give the data of the patents after licensing.

**Opportunities For Improvement:**

If the authors could give the final version of the corresponding licensed patents, the dataset would be more versatile and valuable.
It would be perfect to collect some similar patents from the patent offices of other countries.

**Relation To Prior Work:**

Yes, this paper explicitly discusses how this work differs from previous contributions.

**Summary And Contributions:**

This paper presents a large-scale, well-structured, multi-purpose patent application dataset (HUPD). Unlike other patent datasets, HUPD contains the version of patent applications filed by inventors rather than the final version of issued patents. The dataset also contains rich structured metadata. In addition the paper introduces a new task, namely patent acceptance prediction. By using the data in HUPD, researchers can conduct prediction studies of patent acceptability and explore the success rate of patent applications.

---

> ### Author Response · Authors · 2023-08-10
> **Response to Reviewer sQe3**
>
> Thank you for your positive comments and helpful feedback!
>
> With regard to your opportunities for improvement, our current version of the dataset focuses exclusively on patent applications and the application metadata, but if researchers request it, we could also add the final versions of the patents. We would add this data after we finish adding all  the recent patent application data (e.g. from the years 2019-2023), which is what we are currently working on adding now that the more recent metadata has been provided by the USPTO.
>
> We also agree that it would be quite interesting to be able to combine our dataset with another dataset of patents from a patent office from another country/region (e.g. the EU). This is out of the scope of our current work, but it could be an interesting future extension of the work! For example, it would be interesting to compare the patent literature from different patent offices, find corresponding patents, and find areas of innovation where one patent office has more patent activity than the other.
>
> Thank you very much for your thoughtful comments!

---

### Official Review · Reviewer_DL8S · 2023-07-08
**Review of The Harvard USPTO Patent Dataset**

**Rating:** 8
**Confidence:** 5

**Strengths:**

- A large dataset of USPTO patents is made available
- The dataset is downloadable
- It is larger than other patent datasets
- Some patent background is explained
- Baseline results for four of the tasks in which the dataset can be used are provided
- Further tasks are discussed
- Limitations, biases, and ethics are discussed
- The dataset is foreseen to be useful by both the NLP community and the IP community.

**Additional Feedback:**

There are some references to figures and tables that are not in the paper. In Section 5 there is a missing section reference number.

**Clarity:**

The paper is well written.

It is not explained what "USPTO continuity data files" are.

**Correctness:**

One of the tasks for which baseline results are provided is Patent Acceptance Classification. In the paper it is formulated as a classification task - given the text of the patent, let a classifier predict if the patent will be accepted or not. This is an over-simplicication of a very complex task involving hours or more of work (formulating search queries, analysing patent text in detail, etc.) by an expert in the subject matter of the patent under consideration. This limitation should at least be mentioned in the paper.

**Documentation:**

Documentation is available on the dataset website.

**Ethics:**

No ethical concern expected.

**Limitations:**

The authors provide a detailed discussion of the limitations.

**Opportunities For Improvement:**

- It is not clear how and why the time window of 2004-2018 for included patents was chosen
- The dataset only contains patents from the USPTO, which limits its geographical applicability to the US (listed as a limitation in the paper)
- The dataset contains only patents in English, which means that it cannot be used for multilingual tasks (listed as a limitation in the paper)
- The dataset does not contain patent drawings (listed as a limitation in the paper)
- Formulating Patent Acceptance Prediction as a classification problem is an over-simplification of a very complex task


**Relation To Prior Work:**

The relation to prior work is acceptably covered. Related patent datasets are compared.

**Summary And Contributions:**

The paper presents a large-scale consistently-structured public corpus of 4.5 million English-language USPTO patents between 2004 and 2018. It focuses on patent applications, not granted patents, which opens some interesting applications.

The paper discusses multiple tasks in which the dataset could be used. Baseline results are provided for four of the tasks (Patent Acceptance Prediction, Automated IPC classification, language modelling and abstractive summarisation).

---

> ### Author Response · Authors · 2023-08-10
> **Response to Reviewer DL8S**
>
> We thank the reviewer for the constructive feedback and valuable suggestions! And we are very pleased to hear that the reviewer has found our dataset useful and thoroughly documented. In what follows, we would like to provide further clarifications about our dataset based on their questions and comments:
>
> **Regarding USPTO Continuity Data Files**:
>
> Our apologies for the oversight in our explanation. Continuity data in patent applications is a way to connect an original patent application with its continuation applications. The USPTO makes this information available in two separate files on the PatEx website. The file named "continuity_parents" includes information about the initial parent application, while "continuity_children" details the following continuation applications that stem from it. These two files together are referred to as the USPTO continuity files throughout the paper.
>
> **Figures, Tables, and Sections in the Appendix**:
>
> Due to page constraints, we were unable to include all relevant figures, tables, and statistics in the main part of the paper, so we moved some of this content to the Appendix. Should we be granted an additional page upon acceptance, we intend to transfer some figures and tables back to the main body. Additionally, we would like to clarify that the missing reference in line L222 should have been directed to Section H in the Appendix—we apologize for any confusion caused by this error.
>
> **The Complexity of the Patent Acceptance Task**:
>
> Your feedback on the complexity of the patent acceptance prediction (patent decision classification) task is on point and aligns with our views.  We acknowledge that our first description may have oversimplified the task, and we are really thankful for the constructive critique and insightful feedback. In our revised version, we will elaborate on these complexities to ensure that our readers have an understanding of the multifaceted nature of this task.
>
> Thank you very much for all your comments!

---

### Official Review · Reviewer_Gojv · 2023-07-20
**A good paper, well written, presenting comprehensive data and extensive experiments.**

**Rating:** 7
**Confidence:** 3
**Correctness:** Yes
**Clarity:** Yes

**Strengths:**

1. This paper is well-written, providing a clear and comprehensive overview of the related work.
2. The datasets include the rejected patents, which can be used to predict the acceptance likelihood of a patent application.
3. The experimental results and visualization in the attachment are extensive. The experiment on the evolution of patent applications is interesting.


**Additional Feedback:**

See above

**Documentation:**

Yes

**Opportunities For Improvement:**

1. The authors claim the largest patent data, including 4.5 million patent applications. However, there seems to be no significant disparity in magnitude when compared to existing datasets. Also, this collection still is a small fraction of the real-world patents.
2. The downstream task on this dataset is somehow limited and old-school. It would be intriguing to explore more practical tasks, such as detecting plagiarized patents.
3. The results obtained for Patent Acceptance Prediction and the top-1 score for Subject Classification do not appear to be very high. It would be valuable if the authors could provide an explanation for the observed performance. What are the challenges that make these tasks particularly difficult?


**Relation To Prior Work:**

Yes

**Summary And Contributions:**

This paper provides a patent dataset, including 4.5 million patent documents, including rejected patents, and rich structured metadata. The datasets are used for four tasks: classification, language modeling, summarization, and acceptance prediction. This proposed data may have the potential to become a benchmark for various text classification or patent-oriented intelligent applications, making a valuable contribution to the community.

---

> ### Author Response · Authors · 2023-08-10
> **Response to Reviewer Gojv**
>
> Thank you for your positive review and for your work reviewing our dataset! We are really delighted to learn that the reviewer has found our analyses and documentation extensive and thorough.
>
> With regard to your opportunities for improvement:
>
> 1. It is true that this dataset contains only a fraction of all real-world patents. Nonetheless, this dataset is larger than prior datasets, and perhaps even more importantly, it contains rich metadata that previous datasets lacked. This metadata enables researchers to tackle new tasks that were previously outside the scope of large-scale legal NLP research. Even though the dataset does not contain all patent applications submitted to all patent offices, these elements of the dataset are  valuable to the NLP and legal/IP communities.
>
> 2. Thank you for mentioning this! Detecting plagiarism in patent applications is indeed  an important topic which our dataset could absolutely be used to investigate. The purpose of our experiments was to establish baselines for the more traditional (“old-school”) tasks such as summarization because we believe it is important for new datasets to be accompanied by standard baselines (with which most people in the research community are already familiar). That being said, we would be delighted if future research used our dataset to study patent plagiarism detection.
>
>     * Additionally, we would note that in the future, large language models will be inevitably by used to help write and submit patents; it could be interesting to investigate the issue of human-versus-LLM-written patents using the dataset (because we will update the dataset to contain patent applications from 2019-2023 shortly). These issues are closely related to the issue of plagiarized patents.
>
> 3. The accuracy for patent acceptance prediction is not very high because of the difficulty in predicting whether or not a patent will be accepted and assessing its claims. This is true even for highly experienced patent attorneys, whose job is in part trying to understand these factors, and who may take several months to evaluate the claims.
>
>     * The process of determining whether an invention is patentable involves a complex analysis of various factors. For example, the assessment of non-obviousness of a patent can be influenced by many factors, such as the state of the art in the relevant field and the differences between the claimed invention and the prior art. This assessment may require a large amount of expertise and research in a given field, and despite this it may still vary from one examiner to another. We can discuss these challenges more in the text.
>
>     * For the subject classification task, the primary challenge is that the subjects are very fine-grained: there are 637 classes (IPC codes at the subclass level). For example, D04G is “Making Nets By Knotting Of Filamentary Material” , D04H is “Making Textile Fabrics, e.g. From Fibres Or Filamentary Material”, D05B is “Sewing; Embroidering; Tufting”, and D05C is “Embroidering; Tufting”. Given the difficulty of this task, the top-1 scores of our best-performing models are quite strong.
>
> Thank you very much for your thoughtful comments!

---

### Official Review · Reviewer_4nA3 · 2023-07-22
**A valuable dataset described by a strong paper, with some addressable concerns**

**Rating:** 7
**Confidence:** 5

**Strengths:**

* **S1.** The paper is well written, easy to follow, and quite extensive. The authors do an excellent job at contextualizing their work for NLP/ML practitioners, and the investigations proposed are interesting. Finally, they provide sufficient details to ensure the quality of their experiments and work.

* **S2.** The dataset improves upon the state-of-the-art on a variety of axes (submitted text vs revised version, metadata, scale), and could prove valuable to the community -- especially for practitioners working at the intersection of law and AI.

* **S3.** The authors conduct an in-depth analysis (in the Appendix) of the biases carried by their dataset.

**Additional Feedback:**

This is overall a valuable dataset described by a strong paper, offering novel use cases for the community. The results presented are a good teaser of what the dataset enables, altough the presentation of the language modeling and summarization tasks could be improved (W1/2). Finally, it's unclear whether this dataset will be updated in the future as new patents are released, or if it would stay in its current state. Accordingly, I rate this work as a **6 (Weak Accept)**. Should my concerns be addressed, I would be willing to increase my score to a 7 (Accept).

**EDIT: following rebuttal/discussion, I have updated my score to a 7 (Accept).**

* **Q1.** (W1.) Could the authors clarify their results on language modeling? It is unclear to me currently where they are presented in the paper and what is their value.

* **Q2.** (W2.) Could the authors provide oracle/lead/random (picking a random sentence from the text to summarize) baselines for the ROUGE scores?

* **Q3.** (W3.1.) Could the authors provide at least a snippet of code enabling users to retrieve the final revised texts, as mentionned in the footnote?

* **Q4.** (W3.2.) Could the authors clarify their choice of cut-off dates? Do they intend to update the dataset in the future?

**Clarity:**

Yes, and the authors contextualize patent law concepts appropriately for unfamiliar readers coming from machine learning.


**Correctness:**

The process described to construct the dataset is sound, and leaves limited room for error. The experiments are extensively documented and provide a good way to demonstrate some of the possibilities opened by the release of this dataset.

**Documentation:**

The authors have adequately documented the dataset through a dataset card, and have also released pipelines, code, and model for experiments allowing for reproducibility.

**Ethics:**

No.

**Limitations:**

The authors adequately discuss limitations of their work throughout the paper, and document extensively the potential biases and negative impact of their work.

**Opportunities For Improvement:**

* **W1. Language modeling task.** The authors discuss a language modeling task, but perplexity scores are not reported in any of the table and/or figures. Furthermore, the authors use masked language modeling for this task; this is an uncommon choice for reporting perplexity, and a causal decoder-only model would probably have been a better choice. However, the masked representations are relevant for the embeddings the authors construct later.

* **W2. Contextualizing ROUGE scores.** To contextualize the ROUGE scores reported, lead/oracle baselines would be valuable -- especially since the authors discuss lead sentences in footnote 15.

* **W3. Comprehensiveness of the dataset.**

	* **W3.1.** In footnote 6, the authors mention the "final" revised text of the patent can be retrieved but that the dataset is distributed without that content. This seems like a drawback for a comprehensive patent dataset.

	* **W3.2.** It's unclear why the cut-off dates have been selected as 2004-2018. Is this due to a format change before/after these dates? It's also not clear if this dataset will be updated in the future, or if this artefact is considered complete.


* **W4. Readibility of figures.** Figure 3 and Figure 5 all present absolute values, which make it difficult to compare relative e.g., percentages of acceptance across categories.

* **W5. Minor nits.**

	* **W5.1.** l27 "it [patent analysis] has yet to be systematically studied by the ML and NLP communities" --> l120 "there has been an interest in applying language models to patents". The claim on l27 seems unjustified given the further (extensive) discussion of the litterature at the interface of patent law and NLP.

	* **W5.2.** Altough it is mentionned in the Appendix, the USPTO dataset from The Pile (Gao et al., 2020) would be a welcome addition to Table 1. It is quite different from the other datasets mentionned, as it explicitely targets language modeling. For practitionners working on large language models, it is probably the first patent dataset that comes to mind.

	* **W5.3.** Table 2, colum 3 is "Avg # Tokens" but these numbers are later referred as word counts. If these are not words but tokens, the tokenizer used should be specified.

	* **W5.4.** l167 "upon publication" --> that content is already in the Appendix.

	* **W5.5.** The glossary could be referenced to earlier in the introduction to direct readers to it.

**Relation To Prior Work:**

Yes, the authors adequately cite and discuss previous work.

**Summary And Contributions:**

This paper introduces HUPD, a large-scale structured dataset of patent applications to the USPTO from 2004 to 2018. At variance with previous patent datasets, HUPD includes texts as submitted to the USPTO instead of the final revised versions, as well as complete metadata. This enables novel analyses and NLP applications on top of the dataset, such as acceptance prediction. The authors explore a number of example the tasks enabled by HUPD, from the aforementionned acceptance prediction, to summarization or IPC code (subject) classification.

---

> ### Author Response · Authors · 2023-08-10
> **Response to Reviewer 4nA3**
>
> Thank you for your comprehensive review and for all the effort you put into reviewing our work! Your opportunities for improvement are very helpful and we will use them to improve our paper.
>
> **W1.** We acknowledge that masked language modeling is less popular than causal language modeling. As the reviewer noted, we picked masked language modeling in order to obtain more useful embeddings for the other downstream tasks. In our first draft of the paper, we included perplexity numbers for masked language modeling (1.56 PPL), but we removed these in subsequent drafts because our readers did not know how to interpret them (because most readers are only used to standard causal perplexity numbers). If it would be helpful, we are happy to train a causal language model on the data and report PPL numbers. For example, we could fine-tune GPT-2 or LLaMa (or potentially an 8K-context-length version of LLaMa) on the dataset and report the results.
>
> **W2.** We entirely agree with the reviewer, and we will add lead/oracle ROUGE baselines to the tables in the final version of the paper.
>
> **W3.1.** We would like to apologize for not outlining how to get the final version of a given patent application.  Here is how to obtain the final version of a patent application: (i) Extract the application's title and inventor information. (ii) Use this to search and filter the metadata frame. (iii) Identify all related patent applications within the family. (iv) To find the final version, sort these applications by publication date.  (v) Select the last row in the data frame. (vi) To retrieve the text, call the data loader with the application number found previously. Let us note that we plan to provide a section, along with relevant code,  in our dataset tutorial to help users with this.
>
> **W3.2.** For the cutoff dates, we begin with 2004 because this is the earliest year that cleanly distributed full-text records from the USPTO Bulk Data Storage System in the XML format we are parsing (as described in Section 4) are available. We ended with 2018 because when we began the project, we originally intended to use the most recent years as completely-held-out test data (e.g., for researchers studying language drift). *We will be updating the dataset to include this new data (from 2019-2023) in the fall.*
>
> **W.4.** Thank you for helping improve the readability of Figures 3 and 5. We will add relative value in parentheses next to the absolute values.
>
> **W5.1.** We will soften the claim on L27.
>
> **W5.2.** Very good point; we will add the USPTO subset of the Pile to Table 1.
>
> **W5.3.** These are tokens with the GPT-2 tokenizer. We will clarify this in the caption and the main text.
>
> **W5.4.** We will remove “upon publication”
>
> **W5.5.** We will note in the introduction that there is a glossary for the reader’s reference.
>
> Thank you very much for your thoughtful comments!

---

> > ### Comment · Reviewer_4nA3 · 2023-08-17
> > **Answer to rebuttal**
> >
> > First I would like to thank the authors for providing a rebuttal to each reviewer. The authors have adequately answered to my concerns, and I believe that the paper should be stronger with the additional clarifications. Accordingly, I am revising my score to an **Accept (7)**.
> >
> > > For example, we could fine-tune GPT-2 or LLaMa (or potentially an 8K-context-length version of LLaMa) on the dataset and report the results
> >
> > I do not think this is strictly necessary, as perplexity scores can be difficult to interpret or to contextualise across datasets. However, I think it would be valuable to clarify the main text regarding the use of masked language modeling, and that such models would be best suited to classification/embeddings task.

---

### Official Review · Reviewer_YoRT · 2023-07-26
**Harvard USPTO dataset encompasses the entire lifecycle of a patent application**

**Rating:** 7
**Confidence:** 4
**Clarity:** The paper is very clear and well writ…

**Strengths:**


The present work curates, for the first time, public documents that encompass the entire lifecycle of a US patent application. This leads to a new patent acceptance prediction task and LLM benchmarks on this and several other key tasks relating to the patent examination process. This submission appears to be a thorough, technically strong, well-written and user-friendly contribution towards datasets and benchmarks for evaluating LLMs in the legal domain, an important research direction.

**Additional Feedback:**

N/A

**Correctness:**


The results in the paper seem to be sound and correct.

**Documentation:**

The dataset is released under a noncommercial CC license, and is available on HuggingFace, while the code and pretrained models are released under the MIT license on GitHub. The repository and dataset appear to be very well documented and well organized in accordance with best practice.

**Limitations:**

*As made clear by the authors, the models are evaluated on patent acceptance prediction and subject classification using only short sections of each patent application, namely abstracts and claims, but not the longer background, summary or description sections.  The authors note that there was no significant difference between NB classifiers and BERT-based models in terms of accuracy of patent acceptance prediction, and posit that models for longer text may improve accuracy.  (Regarding this point, I have made some suggestions below in relation to prior work.)

*I believe applicants may make an election to keep the utility patent applications confidential until and *unless* they are granted by making a special nonpublication request.  It may be worth double checking this, and if this is correct clarifying this in the paper - this aspect of the patent disclosure process may introduce a potential bias towards successful applications in the publicly available materials.

**Opportunities For Improvement:**


It may be worth clarifying whether the raw patent data from USPTO contains any formatting information, such as bold font or other special formatting of sections and subsections. If this formatting information is available, it would allow multimodal models to better process/create hierarchical representation of the long sections in a patent application, such as the description and possibly also the background and summary.

Further to the previous paragraph, on the abstractive evaluation task, the T5 model is evaluated using the descriptions and claims in patent applications, which likely exceed the model's maximum token limit based on the dataset statistics. It may be worth clarifying whether any special steps, like chunking of the text, were taken to overcome this limit.

**Relation To Prior Work:**

I have a suggestion in re Section 6 mentioning that models in future studies may include the longer sections of the patent applications, such as the background and description, and Appendix G.1 on Long Sequence Modeling. Here a few references dealing with long-sequence modeling in the closely related context of legal contracts may help make progress towards incorporating the longer sections of the patent applications into language models.
*Li et al, Don't Use a Cannon to Kill a Fly: An Efficient Cascading Pipeline for Long Documents;
*Hegel et al., The law of large documents: Understanding the structure of legal contracts using visual cues. In Document Intelligence Workshop at KDD, 2021; and
*Rao et al., MarkupMnA: Markup-Based Segmentation of M&A Agreements.

**Summary And Contributions:**

The submission describes the US patent examination process and introduces a dataset containing over 4 million patent documents, including applications and related documents with rich structured metadata. These documents concern patent applications relating to inventions (called "utility" patents) to the US Patent and Trademark Office during 2004-18. The dataset contains the original inventor-submitted versions, including rejected and pending applications as well as the granted versions, along with metadata and fields, including claims, background, filing data and examiner information, and labels with the decision status. The paper introduces a new patent acceptance prediction task in addition to revisiting the automated subject classification and abstractive summarization tasks introduced previously in the literature. It evaluates naive Bayes (NB), logistic regression and BERT-based models on the first two tasks and the T5 model on the last task.  The paper also releases a masked BERT-style language model pretrained on the dataset.

---

> ### Author Response · Authors · 2023-08-10
> **Response to Reviewer YoRT**
>
> Thank you for your positive comments and insightful opportunities for improvement!
>
> With regard to richer formatting information, such as bolding, the dataset currently does not include this information, as the sections and subsections are represented as raw text. However, if this rich formatting would be helpful, we would likely be able to provide it. Currently, we plan to add a section on our website asking users to send us an e-mail if they are interested in the richly-formatted data; if the community is interested, we will prepare this data and release it with the second version of the dataset, alongside the text-only version.
>
> With regard to the abstractive summarization task, it is true that the lengths of the claims sections of some patents exceed the maximum token length of the T5 model during training. We would note, however, that because the T5 model is an encoder-decoder model with relative positional embeddings, it is possible to use a longer sequence length during inference-time. As a result, we were able to use a longer sequence length for evaluation than one might initially expect (up to 4096 tokens, after which we were limited by GPU memory). When the claims exceeded this length, we simply truncated them. The reviewer is absolutely correct that in the future, it would be interesting and worthwhile to try other strategies for processing longer documents, such as chunking, retrieval-based methods, and models designed for long context lengths.
>
> Finally, thank you for your helpful references with regard to models with longer context lengths! We will be adding all the references you list (Li et al., 2023; Hegel et al. 2021; Rao et al., 2023) to our discussion of the related literature.
>
> Thank you very much for your thoughtful comments!

---

### Author Response · Authors · 2023-08-10
**General Response to All Reviewers**

Dear Reviewers,

We are grateful for your efforts in reviewing our paper and thrilled that each of the five reviewers has given it an "Accept" rating.

We will be taking all your insightful feedback, comments, and critiques into consideration to enhance and clarify our work. Please find our individual responses to each of your reviews below.

Thank you once again!

Sincerely yours,

Authors of the HUPD paper.

---

### Decision · Program_Chairs · 2023-09-22

**Decision:**

Accept (Spotlight)

**Comment:**

Reviewers agree that this is a good paper that should be accepted, and that the dataset is a valuable contribution to the community.

Strengths:
- The dataset is an improvement in the state of the art on an important topic, with more information provided than in previous work; and it is large, well-structured, downloadable, and notable includes rejected patents.
- This improved dataset opens up the possibility of new uses for ML, and the authors highlight in particular patent acceptance prediction
- The authors provide in-depth analyses, results, and visualizations.

Opportunities For Improvement:
- There are other potential applications that stem from this dataset that the authors do not cover.
- The dataset is still limited in some ways (language, geography, missing drawings).
- Additional issues have been addressed in author responses and can be updated for a camera-ready.

Limitations:
- Are well described by authors.

Correctness:
- Overall correct, minor concern with the proposed task being overly simple.

Clarity:
- Very well written

Documentation:
- Good

Ethics:
- Minor